# Extent of carbon nitride photocharging controls energetics of hydrogen transfer in photochemical cascade processes

Oleksandr Savateev [1,5] ✉, Karlo Nolkemper[1,2], Thomas D. Kühne[2,3,4], Vitaliy Shvalagin[1], Yevheniia Markushyna[1] & Markus Antonietti [1]

Graphitic carbon nitride is widely studied in organic photoredox catalysis. Reductive quenching of carbon nitride excited state is postulated in many photocatalytic transformations. However, the reactivity of this species in the turn over step is less explored. In this work, we investigate electron and proton transfer from carbon nitride that is photocharged to a various extent, while the negative charge is compensated either by protons or ammonium cations. Strong stabilization of electrons by ammonium cations makes proton-coupled electron transfer uphill, and affords air-stable persistent carbon nitride radicals. In carbon nitrides, which are photocharged to a smaller extent, protons do not stabilize electrons, which results in spontaneous charge transfer to oxidants. Facile proton-coupled electron transfer is a key step in the photocatalytic oxidative-reductive cascade – tetramerization of benzylic amines. The feasibility of proton-coupled electron transfer is modulated by adjusting the extent of carbon nitride photocharging, type of counterion and temperature.

Photocatalysis utilizes electromagnetic radiation in the UV-vis range of the electromagnetic spectrum to overcome an activation energy barrier and to drive uphill reactions. Among many possibilities for classification, photocatalytic processes may be divided into (i) those that are enabled by a single photon—a mechanism includes a single photocatalytic event, and (ii) those that include at least two photocatalytic events, such as combination of photoinduced electron transfer (PET), excited state proton-coupled electron transfer (ES-PCET)[1] or direct hydrogen atom transfer (dHAT)[2] and/or energy transfer (EnT)[3] (Fig. 1a). In cascade photocatalysis, it is not always possible to unambiguously confirm if a second photon is involved[4]. However, once a thermodynamically stable closed-shell and electrically-neutral compound in the ground state is formed, a subsequent photocatalytic event is required to activate such a compound. There are already plenty

examples of cascade processes mediated by molecular photocatalysts, such as (−)-Riboflavin[5], thioxanthen-9-one[6], Ir-polypyridine complexes[7,8], Eosin Y[9], [Mes-Acr]+ClO4−[10], which involve either two sequential EnT events[6–8], two PET events[9,10], or a combination of EnT/PET events in the first and second steps[5]. In semiconductor photocatalysis, a rather well-studied oxidative photocatalytic cascade hydrocarbon-alcohol-aldehyde-carboxylic acid proceeds via combination of ES-PCET (dHAT) and/or PET events[10]. A process such as $CO_2$–$HCOOH$–$H_2CO$–$CH_3OH$–$CH_4$ proceeds via a formaldehyde pathway and a combination of PET and/or ES-PCET (dHAT) events[11].

The chemical structure of reagents and reactions in the above examples are indeed very different (Figure S1). In order to analyze them collectively, in Fig. 1b the change of the formal oxidation number of the atom at the reaction site is plotted along the reaction

[1]Department of Colloid Chemistry, Max Planck Institute of Colloids and Interfaces, Am Mühlenberg 1, 14476 Potsdam, Germany. [2]Dynamics of Condensed Matter and Center for Sustainable System Design, Chair of Theoretical Chemistry, University of Paderborn, Warburger Str. 100, D-33098 Paderborn, Germany. [3]Center for Advanced Systems Understanding (CASUS) and Helmholtz-Zentrum Dresden-Rossendorf, Untermarkt 20, D-02826 Görlitz, Germany. [4]Institute of Artificial Intelligence, Chair of Computational System Sciences, Technische Universität Dresden, 01187 Dresden, Germany. [5]Present address: Department of Chemistry, The Chinese University of Hong Kong, Shatin, New Territories, Hong Kong, China. ✉e-mail: oleksandr.savatieiev@mpikg.mpg.de

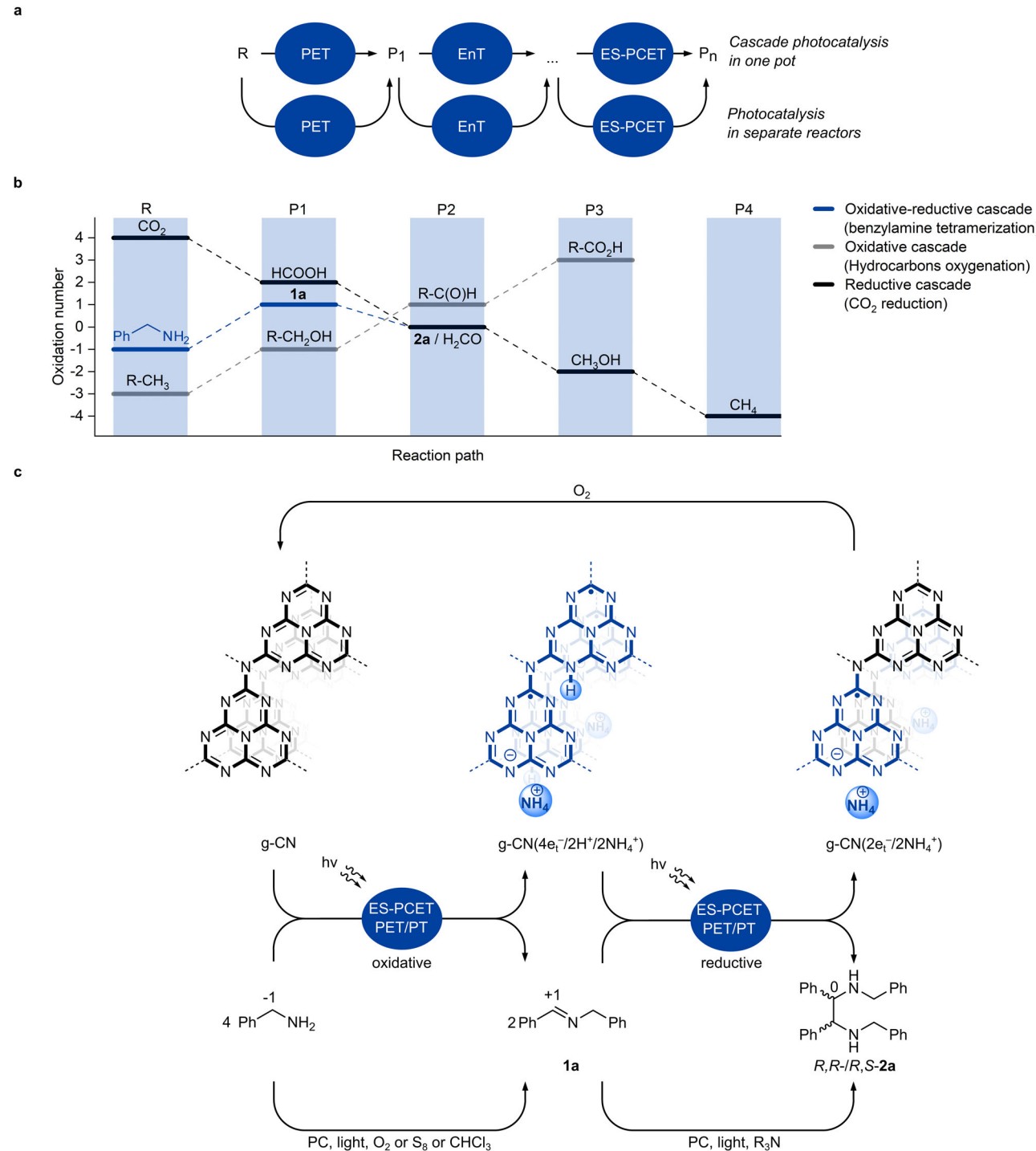

**Fig. 1 | Cascade photochemical processes. a** Schematic representation of a cascade photocatalysis in one pot and each photocatalytic reaction performed in separate reactors. PET—photoinduced electron transfer. EnT—energy transfer. ES-PCET—excited state proton-coupled electron transfer. **b** Change of the formal oxidation number of the reaction site in cascade processes. R—reagent. P1–4,$P_n$—product 1–4, n. **c** An oxidative-reductive cascade process that is based on the transfer of electrons and protons and illustrated by tetramerization of benzylamine. $e_t^-$—trapped electron, H$^+$—proton, PC—photocatalyst, PT—proton transfer.

coordinate. While in redox neutral cascades, in particular those that proceed exclusively via EnT, formal oxidation number does not change[6–8], in oxidative cascades–increases[9,10], and in reductive–decreases[11]. Such an observation stems from the fact that it is challenging to switch between, for instance, oxidative (typically in the presence $O_2$) to reductive conditions (in the presence of tertiary amines) and vice versa in the same cascade. The principle used to create Fig. 1b outlines a type of cascade processes, which demonstrates a maximum or a minimum on the plot and is referred to as an

oxidative-reductive cascade. Such cascades are based on the transfer of atoms, among which the most ubiquitous is hydrogen.

Tetramerization of benzylic amines is an example of a cascade process. Individual steps were reported separately (Fig. 1c). Photocatalytic coupling of benzylamine to imine requires oxidants, $O_2$[12], $S_8$[13], or CHCl$_3$[14]. On the other hand, *aza*-pinacol coupling of imines under reductive conditions is mediated by CdS[15], Ir$^{III}$-polypyridine complexes[16], and transition metal free organic dyes, such as diphenyldibenzocarbazole[17], *N*-phenylphenothiazine[18] and perylene[19].

One of the greatest advantages of cascade photocatalysis is that it merges several reactions into a single process, which overall is more resource-efficient. In order to implement a photochemical oxidative-reductive cascade process, such as tetramerization of benzylamine, a semiconductor, exemplified by heptazine-based graphitic carbon nitride (g-CN) in Fig. 1c, must first enable an oxidative event and at the same time temporarily store one equivalent of electron/proton couples, $e_t^-/H^+$. The subscript stands for 'trapped' electrons to emphasize that the potential of such electrons is different from that of the photogenerated "hot" electrons. Secondly, the photocharged semiconductor must bind $e_t^-/H^+$ only weakly in order to transfer them to the imine in the second step. By contrast strong binding of $e_t^-/H^+$ would terminate the cascade process at imine **1a**. Given that a semiconductor is required in a quasi-stoichiometric quantity versus benzylamine to store temporarily an equimolar amount of $e_t^-/H^+$ pairs, we deliberately omit the term 'catalysis', but refer to benzylamine tetramerization as a 'cascade process'. Nevertheless, a photocharged semiconductor may be recovered upon exposure to $O_2$ or other oxidant.

In this work, benzylamine tetramerization that is enabled by g-CNs, such as mesoporous graphitic carbon nitride (mpg-CN), sodium poly(heptazine imide) (Na-PHI), and visible light, is chosen as a model oxidative-reductive cascade process. The objectives of the present study are: (1) identify preferential facets of g-CN materials at which charge-compensating cations are stored; (2) investigate the influence of the nature of charge-compensating cations, $H^+$ vs. $NH_4^+$, on reactivity of $e_t^-$; (3) establish the role of micro- and meso-porosity in storage of electrons and charge-compensating cations.

## Results

### Tetramerization of benzylamine

Illumination of a thoroughly degassed mixture of benzylamine and a semiconductor (g-CN, $TiO_2$, CdS) in MeCN with photons of suitable wavelength to ensure electron excitation across the band gap gave imine **1a** and a mixture of diastereomers *R,S*-**2a** and *R,R*-**2a**. The diastereometric ratio (d.r.) was determined from $^1$H NMR spectra of the reaction mixture[20]. While the full data set is given in Supplementary Table 1, we comment on general trends. Conversion of benzylamine, the yield of **1a** and the combined yield of *R,S*- and *R,R*-**2a** diastereomers versus the mass of carbon nitride semiconductor, Na-PHI and mpg-CN, are plotted in Fig. 2. In most cases, after 24 h **2a** was the major product, indicating the feasibility of temporary storage of hydrogen in semiconductors and transfer to the imine **1a**. In the experiments with the same mass of semiconductor, mpg-CN-8nm-193 (average pore diameter 8 nm and $S_{SA} = 193$ m$^2$ g$^{-1}$), gave higher conversion and yield when the reaction was conducted at 80 °C compared to room temperature and compared to Na-PHI (specific surface area, SSA, 1 m$^2$ g$^{-1}$). The extent of conversion depends on (i) the amount of Na-PHI and mpg-CN-8nm-193 and (ii) surface area of the semiconductor (Supplementary Discussion 1). For mpg-CN-8nm-193, complete conversion was achieved at 160 mg of the semiconductor, while for Na-PHI complete conversion was not achieved. In all experiments, the yield of *R,S*- and *R,R*-**2a** remained below 60%. H-PHI, obtained from Na-PHI upon Na$^+$ exchange with protons[21], gave *R,S*-**2a** and *R,R*-**2a** in significantly lower combined yield, 8% (entry 31, Supplementary Table 1). Control experiments confirmed that mpg-CN and anaerobic conditions were essential to obtain **2a** (Supplementary Table 2). The presence of $O_2$ terminated the cascade process at the first step—**1a** was obtained in 28 ± 2% yield and **2a** did not form. Benzylamine was converted to neither **1a** nor **2a** in the absence of a semiconductor and in the dark. Overall, benzylic amines ring-substituted with fluorine atoms and CF$_3$-groups also give ethanediamines. On the other hand, in the case of benzylic amines carrying electron-donating groups in the aromatic ring and sterically encumbered 1-phenylethanamine, the cascade process is terminated at the step of the corresponding imine synthesis.

After illumination was stopped and before exposure of the reaction mixture to air all semiconductors underwent an obvious color change, which is attributed to the accumulation of electrons[22]. Among carbon nitrides, the most pronounced color change is observed for PHIs[23]. In the DRUV-vis spectra (Fig. 3a, b) mpg-CN shows an absorption band that stretches up to 800 nm in addition to widening of the band gap by ~10 nm (8 meV) attributed to the Moss-Burstein shift—blue-shift of the onset of absorption at ~450 nm. In a series of parallel experiments performed under the conditions identical to that in Fig. 2, the specific concentrations of electrons, δ, in Na-PHI ($S_{SA} = 1$ m$^2$ g$^{-1}$) and mpg-CN was determined by quenching the reaction mixture with methyl viologen bis(hexachlorophosphate), $MV_2^+$ $2PF_6^-$ (reduction $E_{p1/2} = -0.45$ V vs. SCE in MeCN, Fig. 3c, Supplementary Tables 3 and 4). While the concentration of benzylamine in the reaction mixture was kept the same, we observed in all cases a strong dependence of δ on the mass of the semiconductor taken for the experiment (Supplementary Discussion 2). The higher the excess of benzylamine per carbon nitride mass, the larger is δ. Ionic carbon nitride, Na-PHI, accumulates a 3–7 times higher number of electrons per mass unit compared to mpg-CN. Although using a slightly different concentration of benzylamine, 0.1 M herein versus 0.05 M used earlier[24], δ measured for Na-PHI is comparable to that of K-PHI ($m = 5$ mg). H-PHI shows values comparable to mpg-CN δ, obtained for the same mass of the material, which illustrates a transition from an ionic structure in Na-PHI to covalent as in mpg-CN upon substitution of Na$^+$ with protons. Coupling of two benzylamine molecules is accompanied by the formation of $NH_4^+$. Substitution of $H^+$ by alkali metal ion occurs in aqueous environment at pH > 7[21,25,26]. Benzylamine tetramerization was conducted in less polar solvent, MeCN, without adjusting the pH. Therefore, we speculate that under such conditions spontaneous $H^+$-to-$NH_4^+$ ion exchange in H-PHI does not occur. However, given that $NH_4^+$ serves as the counter ion to the stored electron (see below), such ion exchange might proceed via material photocharging. Na-PHI shows systematically lower δ values, when photocharging is conducted at 80 °C suggesting that it is an endergonic process (see the results of DFT calculations below and Supplementary Discussion 3). By correlating δ values (Fig. 3c) with Na content in Na-PHI that was recovered from benzylamine tetramerization reaction mixture (Fig. 2), we found that storage of one electron in the material is accompanied by extraction of 7 and 25 Na$^+$ ions when the reaction is conducted at room temperature and 80 °C (Supplementary Discussion 4).

### Aza-pinacol coupling of imine 1a

In order to study only a second step of the cascade reaction, we used imine **1a** as the substrate and extended the scope of carbon nitride semiconductors to mpg-CN-6nm-204 and mpg-CN-17nm-171 having the average pore diameters of 6 and 17 nm and $S_{SA}$ of 204 and 171 m$^2$ g$^{-1}$, respectively. NH$_4$COOH and tertiary amines, Me$_3$N, Et$_3$N, $^i$Pr$_2$NEt, $^n$Bu$_3$N, and $^n$Oct$_3$N, were used as the source of electrons and protons to generate the α-aminoalkyl radical from the imine **1a**. While a full dataset is given in Supplementary Tables 5–12, below we highlight some of our results (Table 1).

*Aza*-pinacol coupling proceeds efficiently using only 5 mg of mpg-CN-8nm-193 (Table 1, entry 1), which enables us to call this second step of the cascade process as catalytic. It also confirms earlier findings using other classes of photoredox catalysts[15–19]. However, most of the screening experimental conditions were performed with 20 mg of semiconductors in order to correlate the results of the first and the second step of the cascade process. Combination of mpg-CN with either $^i$Pr$_2$NEt (entry 2), Et$_3$N (entry 3) or $^n$Bu$_3$N (Supplementary Table 6, entry 4) gave *R,S*- and *R,R*-**2a** in a combined yield 65–68%. Using NH$_4$COOH in anhydrous MeCN and MeCN/water (3:1 vol.) gave **2a** with 16 and 4% yield, respectively, while conversion of **1a** remained

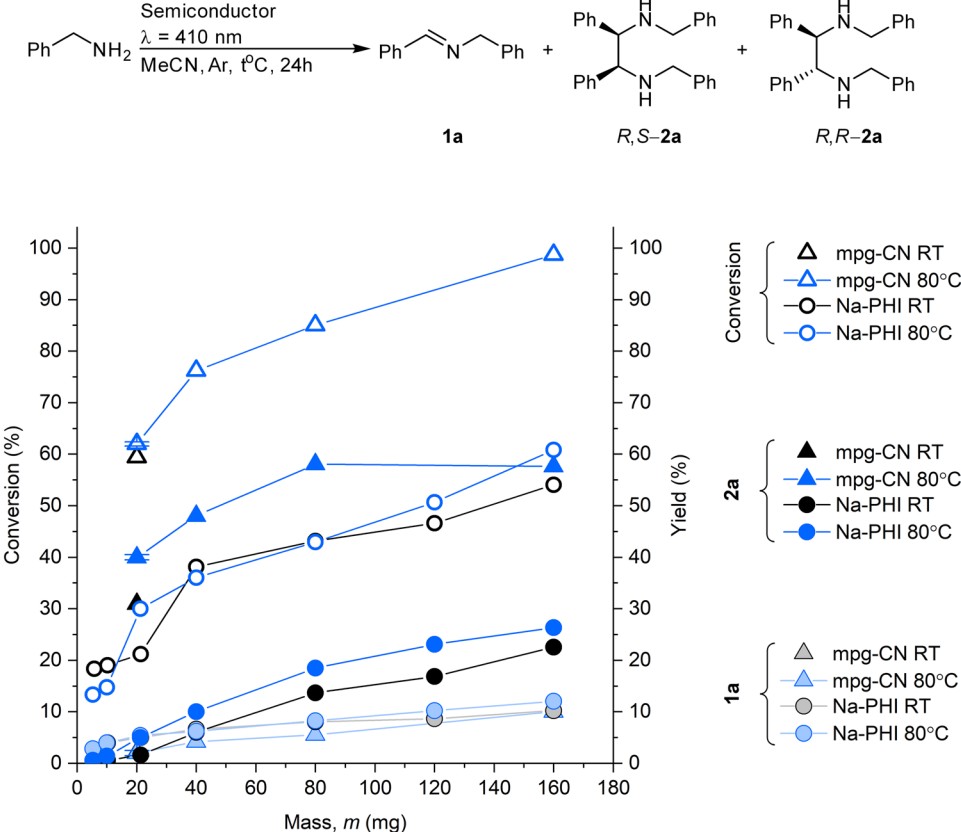

**Fig. 2 | Dependence of benzylamine conversion and yield of imine 1a, *R,S*- and *R,R*-2a on mass of semiconductors and reaction temperature.** Filled symbols correspond to yield, empty—to conversion. Gray/black colors correspond to experiments conducted at room temperature (RT), blue—at 80 °C. Conditions: benzylamine (0.2 mmol), MeCN (2 mL), Ar, 24 h, light 410 nm ($39 \pm 8$ mW cm$^{-2}$). Error bars denote standard deviation ($n = 3$, experiments conducted in parallel). Source data are provided as a Source data file.

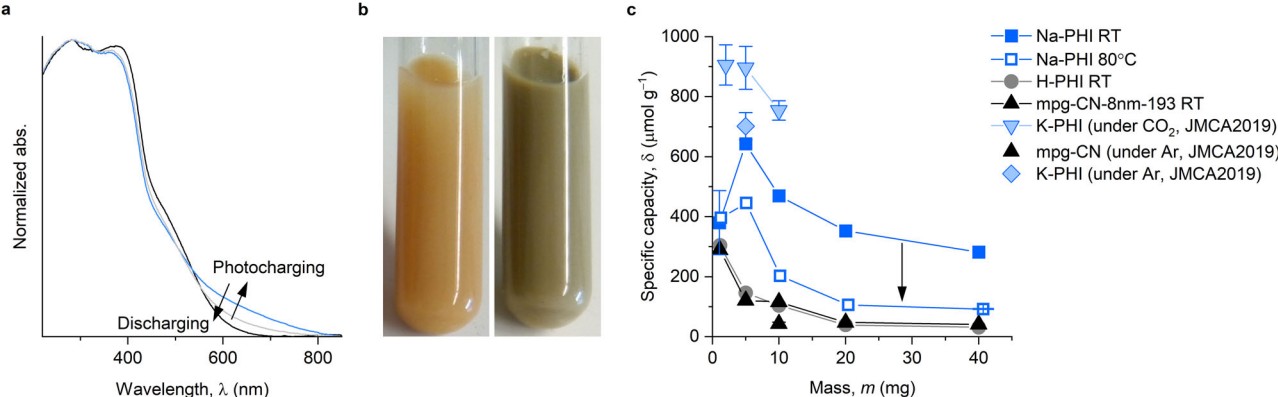

**Fig. 3 | Photocharging of g-CN semiconductors. a** UV-vis absorption spectra of mpg-CN dispersion in $^{i}$Pr$_2$NEt (0.1 M) solution in MeCN prior to irradiation with light (black), after irradiation (blue) and after exposure of photocharged mpg-CN to air (gray). Source data are provided as a Source data file. **b** Photographs of the reaction mixture (mpg-CN-8nm-193 20 mg) before light irradiation and after irradiation at 465 nm for 24 h before exposure to air. **c** Dependence of δ (avg ± std, $n = 3$) on mass of the semiconductor photocharged at room temperature (RT) and 80 °C. Deviation of the data point (avg ± std, $n = 5$) obtained for Na-PHI (1 mg) from the trend is due to low optical density of the reaction mixture. Source data are provided as a Source data file.

low (entries 4,5). With tertiary amines, Na-PHI gave **2a** with lower yield, 46% (entry 6). However, when NH$_4$COOH in anhydrous MeCN or MeCN/water (3:1 vol.) was employed, the yield of **2a** increased to 61% and 33%, respectively (entries 7, 8). A similar trend was observed for H-PHI (entries 9, 10). Although ionic character was lost upon washing with HCl, in *aza*-pinacol coupling, due to the microporous structure, performance of H-PHI was similar to Na-PHI. Synthetic robustness of the method was supported by the fact that the reaction proceeded without thorough degassing of the reaction mixture because dissolved O$_2$ was consumed at the beginning of the reaction (entries 11, 12). *Aza*-pinacol coupling did not proceed without a semiconductor, electron donor and/or light irradiation (entries 13–16).

By performing *aza*-pinacol coupling of **1a** in CD$_3$CN over various semiconductors, we found that Et$_3$N or $^{n}$Bu$_3$N were converted into

**Table 1 | Selected conditions of *aza*-pinacol coupling screening[a]**

| Entry | SC | Mass (mg) | SED | Conversion of 1a[b] (%) | Combined yield of *R,S*-2a and *R,R*-2a[c] (%) |
|---|---|---|---|---|---|
| 1[d] | mpg-CN-8nm-193 | 5 | $^i$Pr$_2$NEt | 94 | 77 (1:0.54) |
| 2[d] | mpg-CN-8nm-193 | 20 | $^i$Pr$_2$NEt | 100 | 68 (1:0.29) |
| 3[d] | mpg-CN-8nm-193 | 20 | Et$_3$N | 98 | 65 (1:0.77) |
| 4[d] | mpg-CN-8nm-193 | 20 | NH$_4$COOH | 26 | 16 (1:0.60) |
| 5[d,e] | mpg-CN-8nm-193 | 20 | NH$_4$COOH | 44 | 4 (1:0.29) |
| 6[f] | Na-PHI | 20 | $^i$Pr$_2$NEt | 46 | 46 (1:0.84) |
| 7[f] | Na-PHI | 20 | NH$_4$COOH | 83 | 61 (1:0.83) |
| 8[f,e] | Na-PHI | 20 | NH$_4$COOH | 61 | 33 (1:0.50) |
| 9[f] | H-PHI | 20 | $^i$Pr$_2$NEt | 66 | 47 (1:0.48) |
| 10[f] | H-PHI | 20 | NH$_4$COOH | 98 | 74 (1:0.61) |
| 11[d,g] | mpg-CN-8nm-193 | 20 | $^i$Pr$_2$NEt | 99 | 67 (1:0.28) |
| 12[d,h] | mpg-CN-8nm-193 | 20 | $^i$Pr$_2$NEt | 97 | 77 (1:0.46) |
| 13[i] | mpg-CN-8nm-193 | 20 | Et$_3$N | 0 | 0 (–) |
| 14[i,j] | mpg-CN-8nm-193 | 20 | Et$_3$N | 0 | 0 (–) |
| 15[d] | – | – | $^i$Pr$_2$NEt | 0 | 0 (–) |
| 16[d] | mpg-CN-8nm-193 | 20 | – | 0 | 0 (–) |

[a]Conditions: **1a** (0.1 mmol), semiconductor, SED (0.2 mmol), MeCN (2 mL), light, 25 ± 5 °C, 24 h.
[b]Yield and d.r. were determined by $^1$H NMR spectroscopy using 1,3,5-trimethoxybenzene as internal standard.
[c]Ratio between *R,S*-**2a** and *R,R*-**2a** isomers is given in parentheses.
[d]465 nm, 10 ± 2 mW cm$^{-2}$.
[e]Reaction in MeCN/water (3:1) mixture (2 mL).
[f]410 nm, 39 ± 8 mW cm$^{-2}$.
[g]Without reaction mixture degassing.
[h]Without reaction mixture degassing, water (10 equiv.) was added.
[i]In the dark.
[j]At 50 °C.

diethylamine and dibutylamine along with the corresponding aldehydes fragmented from the tertiary amines (Supplementary Tables 13, 14). This observation supports the postulated earlier mechanism of tertiary amines, (RCH$_2$)$_3$N, oxidation that proceeds via formation of the intermediary iminium cation followed by hydrolysis to (RCH$_2$)$_2$NH and RC(O)H by trace amounts of water present in the solvent[27–29].

Similar to benzylamine tetramerization, all semiconductors undergo photocharging. To check whether electrons stored in photocharged carbon nitrides are sufficiently reductive to enable *aza*-pinacol coupling of imine **1a** in the dark, a suspension of mpg-CN or Na-PHI and Et$_3$N in MeCN was irradiated with light for 24 h. After addition of imine **1a** to the photocharged carbon nitride semiconductors the mixture was stirred either at room temperature or at 80 °C in the dark for 24 h (Supplementary Table 15). In these experiments, we did not obtain **2a**, but detected a small amount of dibenzylamine, which suggests transfer of 2e$^-$ to the imine **1a** in a single kinetic step.

Upon exposure of photocharged semiconductors to air, coloration vanished within seconds in all cases except Na-PHI and H-PHI that were photocharged with NH$_4$COOH in aqueous MeCN solution−it required hours to recover the pristine yellow color of the semiconductors (Fig. 4a, Supplementary Discussion 5). Although we have been using quite extensively PHIs in photocatalysis under reductive conditions, we have never observed such long-lived species in air.

The exceptionally high stability of PHIs photocharged in aqueous NH$_4$COOH must be due to a synergy of several factors. Firstly, the formate anion[30] is a much stronger reductant compared to benzylamine[24], tertiary amines, and benzylic alcohols[23,31], due to its lower oxidation potential and the irreversible injection of an electron into the photoexcited PHI upon concomitant evolution of CO$_2$. Secondly, an aqueous environment is likely to facilitate transport of formate anions through Na-PHI micropores into the bulk of the material. Therefore, a higher density of electrons in photocharged Na-PHI is achieved, which is reflected in a broader absorption−the material appears black (Supplementary Discussion 6), instead of blue or the ones having a green tint[32]. In a series, H$^+$ <NH$_4^+$ solvated by MeCN <hydrated NH$_4^+$, the last must be the most thermodynamically stable species. Oxidation of photocharged Na-PHI implies extraction of not only electrons from the material, but also charge-compensating cations. Although higher δ values may be achieved by performing photocharging in the presence of ammonium salts in anhydrous MeCN (see below), in aqueous medium, the electrons stored in Na-PHI are less reactive toward O$_2$. Stronger stabilization of electrons by hydrated NH$_4^+$ compared to H$^+$ explains the lower yield of **2a** when Na-PHI was employed in aqueous NH$_4$COOH solution (entry 7 vs. 8 in Table 1)− PCET becomes more challenging.

To complement the results of carbon nitrides photocharging with benzylamine (Fig. 1c), we extended the scope of electron donors to tertiary amines (Fig. 4b, Supplementary Table 4). Higher δ values were obtained for g-CNs featuring micropores, Na-PHI and H-PHI, compared to mpg-CN, which possesses mesopores. Surprisingly, δ does not depend on the $S_{SA}$ of mpg-CN. Assuming that mpg-CN has the structure of melon-type g-CN[33] and stores e$^-$/H$^+$ only on the surface, the obtained δ values are translated into one electron per 1…30 heptazine

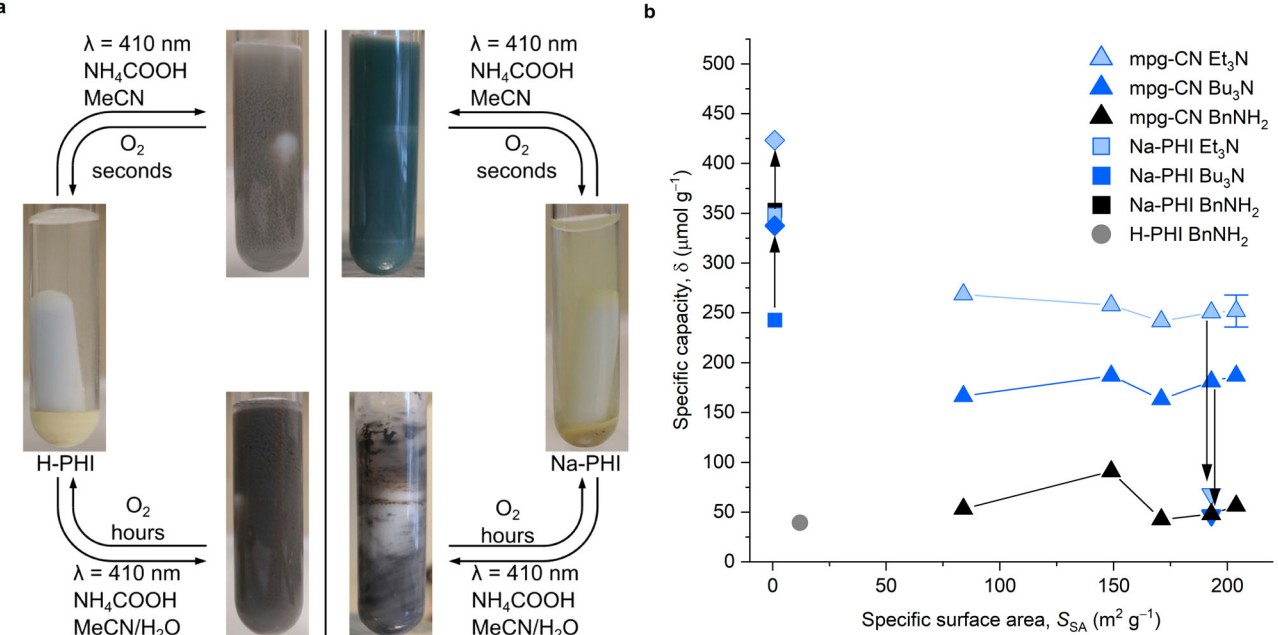

**Fig. 4 | Photocharging of g-CN semiconductors. a** Images of H-PHI and Na-PHI photocharged in MeCN and MeCN/H$_2$O (2:1 vol.) and their stability in air. **b** Dependence of δ (avg ± std, $n = 3$) on $S_{SA}$ of mpg-CN semiconductors determined by quenching with MV$_2^+$ 2PF$_6^-$. Conditions: semiconductor (20 mg), amine (0.2 mmol), MeCN (2 mL), 25 ± 5°, 24 h, 465 nm (10 ± 2 mW cm$^{-2}$, mpg-CN) or 410 nm (39 ± 8 mW cm$^{-2}$, Na-PHI). Arrows indicate shift of data points when photocharging was conducted in the presence of NH$_4$PF$_6$. Source data are provided as a Source data file.

units (Supplementary Table 16, Supplementary Discussion 7). In other words, even in mpg-CN with a $S_{SA}$ as low as 84 m$^2$ g$^{-1}$ the storage of e$^-$/H$^+$ only on the surface is feasible and the surface is not saturated with e$^-$/H$^+$. Higher δ values were obtained for Et$_3$N and $^n$Bu$_3$N compared to benzylamine, which is a result of different pathways of electron donor oxidation. Coupling of two benzylamine molecules to **1a** is accompanied by formation of NH$_4^+$, which serves as a counter ion in the photocharged carbon nitride. NH$_4^+$ stabilize electrons in mpg-CN more efficiently compared to H$^+$, which results in significantly slower rate of MV$^{2+}$ 2PF$_6^-$ reduction. Indeed, when photocharging of mpg-CN was conducted in the presence of R$_3$N and NH$_4$PF$_6$ (anhydrous conditions) we obtained δ values comparable to that of benzylamine. However, addition of NH$_4^+$ in Na-PHI photocharging experiment resulted in 21–39% higher δ values, which confirms that *only in anhydrous* NH$_4^+$-based electrolytes, Na-PHI accumulates a larger number of sufficiently reductive electrons.

However, it is still surprising that although tertiary amines have very similar oxidation potentials ($E_{p1/2}$ = +0.68 and +0.72 V vs SCE in MeCN, Supplementary Table 3, Supplementary Discussion 8), δ values for mpg-CN and Na-PHI obtained in Et$_3$N are 40% higher compared to the reaction in $^n$Bu$_3$N. Such an observation suggests that in mpg-CN electrons and charge-compensating H$^+$ remain localized at the sites where amine oxidation takes place, and they do not equilibrate on the timescale of semiconductor's photocharging followed by quenching with MV$^{2+}$ 2PF$_6^-$, i.e., several hours. Due to the larger diameter of $^n$Bu$_3$N compared to Et$_3$N, the corresponding iminium cation can 'shield' larger areas of carbon nitride from the interaction with electron donor molecules. Given the graphitic-like structure of carbon nitrides with the spacing between the layers of ~0.33 nm, sorption of iminium cations must have the most pronounced effect when occurs at the edge plane (Supplementary Table 17). Taking into account the van der Waals cross-section of the iminium cations derived from Et$_3$N and $^n$Bu$_3$N of 0.30 and 0.44 nm$^2$, respectively, each of them can cover between 0.4–0.9 heptazine units when sorption occurs at the basal plane, but 0.7–2.2 heptazine units—at the edge plane (see SI for calculation details and Supplementary Fig. 2). During mpg-CN synthesis

covalent bonding of cyanamide and its polycondensation products, melem and melon, to a SiO$_2$ surface via abundant NH$_2$-groups (edge plane of the material) is more favorable compared to non-covalent interaction of the π-conjugated system (basal plane of the material) with the silica surface[34]. Therefore, we speculate that hard templating approach gives edge plane-terminated nanocrystalline material.

## A scope of aza-pinacol coupling

A series of ethanediamines **2a-f** was prepared by coupling imines **1a-f** on 3.5 mmol scale in a batch reactor (Fig. 5). At room temperature, the *R,S*-isomers are solids, while the *R,R*-isomers are either liquids, *R,R*-**2a**, **2c-f** or solids with low melting point, *R,R*-**2b**. Due to this feature the isomers were separated by fractional crystallization from methyl *tert*-butylether (MTBE) or pentane. In all cases, slight deviation of d.r. from 1:1 is likely due to a follow up process that selectively affects the *R,R*-isomer (Supplementary Table 18). Imidazolidine is a by-product detected in the reaction mixture (Supplementary Fig. 3). This compound is formed from **2c** and acetaldehyde, which was generated in situ upon triethylamine oxidation[35]. The imine derived from vaniline and 4-trifluoromethylbenzyl amine did not give the corresponding coupling product, likely due to interaction of a phenolic proton with g-CN surface and the subsequent ES-PCET[36,37] rather than tertiary amine. Coupling of **1a** with Zn dust as a chemical reductant in the dark on a comparable scale gave a mixture of dibenzylamine, *R,S*-**2a** and *R,R*-**2a** in 5.25:1:1 ratio and the combined isolated yield of **2a** of 36%[38]. These results point at the advantage of the photochemical approach compared to chemical reduction with Zn dust to reach higher selectivity toward **2a**, 38% vs. 66%. Even if dibenzylamine is formed, mpg-CN converts it into **2a** via a facile dehydrogenation/hydrogenation cascade[15].

## Results of DFT modeling

In order to explain our experimental results we modeled photocharged Na-PHI and H-PHI bearing H$^+$ and NH$_4^+$ as counter ions (Supplementary Discussion 9). Given that the behavior of H-PHI in the experiment with benzylamine is similar to mpg-CN (Fig. 3c) and its

**Fig. 5 | A scope of *aza*-pinacol coupling.** Conditions: imine **1** (3.5 mmol), mpg-CN-8nm-193 (0.7 g), MeCN (70 mL), Et₃N (0.98 mL, 7 mmol), N₂, 25 ± 2 °C, 24 h. ¹H NMR vs. 1,3,5-trimethoxybenzene as internal standard and isolated (in parentheses) yields of ethanediamines. ᵃA mixture of *R,R*-**2b**:*R,S*−**2b** = 1.0:0.15.

crystal structure was reported[39], we also modeled a photocharged state of this material. Simulations reveal a significant difference in energy for adsorption of H⁺ and NH₄⁺ and a different adsorption behavior when comparing H-PHI and Na-PHI.

Structures for Na-PHI and H-PHI are derived from those published by Sahoo et al.[40] by geometrical optimizing cell and structure properties without constraints regarding cell shape or atomic positions. We observe that the strict AA-stacking pattern breaks and the sodium counter ions move towards their energetical optimum in the pores within the g-CN layer (Supplementay Fig. 4).

To model excitation states of the two model structures with static DFT, hydrogen and ammonium radicals were introduced, and adsorption energies observed. To initially identify preferential facets for adsorption a case study for hydrogen has been conducted. Therefore, the interaction of a hydrogen radical at three different positions, labeled as 1, 2, and 3 in Fig. 6a, on periodic systems with 2 heptazine basis units was investigated. The adsorption energy $H_{ads}$ is calculated as described in Eq. (1).

$$H_{ads} = E(PHI + H) - E(PHI) - \frac{1}{2}E(H_2) \quad (1)$$

Note that $E$ is a potential energy and therefore has a negative sign, which yields negative adsorption energies for exothermic process, while positive adsorption energies−endothermic. Desorption energies could be obtained by multiplying the obtained values by −1.

As Fig. 6a shows, only position 3 in H-PHI and position 1 in Na-PHI show exothermic adsorption behavior. In the case of Na-PHI, hydrogens at position 2 and 3 are sterically competing with sodium ions, which are preferentially placed in the corners of the pore, leading to very unlikely, endothermic adsorption energies. For H-PHI, beneficial interactions with nitrogen of the neighboring heptazine, and the proximity of another hydrogen can explain the

energy values for position 3 and 2. Preferential sorption of charge-compensating cations at the edge plane agrees well with the observed earlier migration of photogenerated electrons to the edge plane of PTI/Li⁺Cl⁻ single crystals[41]. Temperature and solvent effects were not included in the model due to the high complexity and demanding computational costs.

Due to the limited pore size, steric effects become more important regarding the adsorption of ammonium, which reduces the amount of potential adsorption sites. These are displayed in the Fig. 6b for H-PHI and for Na-PHI. The first and foremost observation from the figures is that Na-PHI fissures NH₄⁺ into NH₃ and a covalently bonded hydrogen, while no bond breaking appears when NH₄⁺ is adsorbed on H-PHI. This behavior is identical for all studied systems, irrespective of the e⁻ per heptazine density (Fig. 6c, see discussion below). Since $H_{ads}$, which is calculated for this case by adding a potential energy term for ammonia, –$E$(NH₃), to (1), only differs slightly between H- and Na-PHI (Fig. 6b), this divergent adsorbing behavior can, among other explanations, possibly be accountable for the divergent catalytic behavior observed in Fig. 2.

As mentioned above, adsorption energies of H⁺ and NH₄⁺ were calculated for a varying number of heptazine units to investigate the influence of e⁻ charge density, which experimentally corresponds to the mass of added g-CNs in the reaction mixture. The results are plotted in Fig. 6c. A clear tendency of a decrease in adsorption energy with increasing system size is observable, hydrogen adsorption even turns endothermic for systems bearing more than four heptazine units. A possible explanation for this trend is the better delocalization of the electrons within structures with larger π-systems, which decreases the negative charge at the adsorption facet. As adsorption energies decrease, so do desorption energies increase, which lowers the amount of energy necessary to recover g-CN by desorbing e⁻/H⁺ (or e⁻/NH₄⁺).

The results of the DFT modeling indicate that enthalpy of both H⁺ and NH₄⁺ adsorption on H-PHI radical anion depends on the degree of

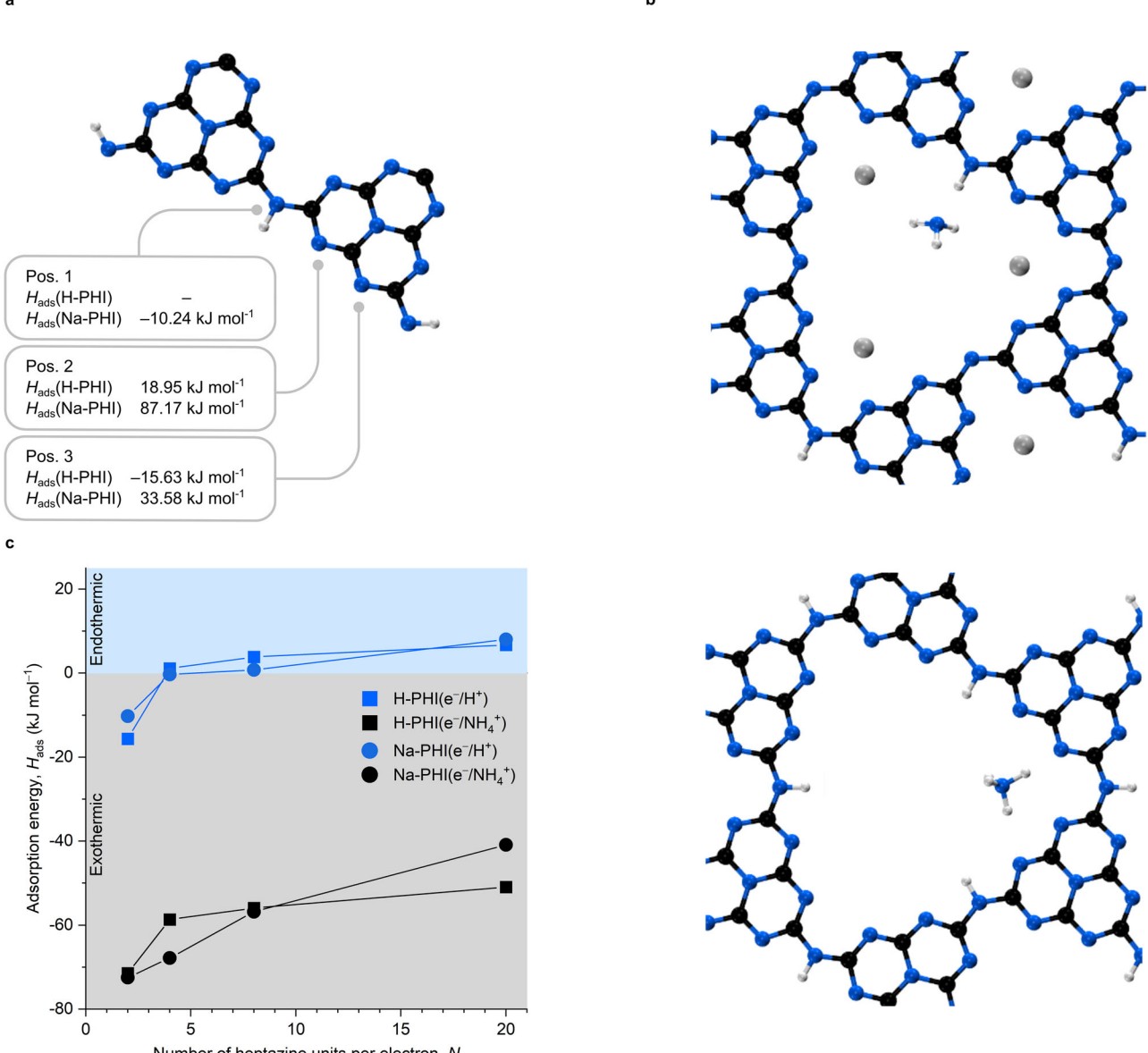

**Fig. 6 | Results of DFT calculations. a** Three in-layer adsorption sites, shown for H-PHI with two heptazine units. **b** Adsorption of ammonium on Na-PHI with $H_{ads} = -72.48$ kJ mol$^{-1}$ (top) and H-PHI with $H_{ads} = -71.46$ kJ mol$^{-1}$ (blue spheres represent nitrogen, gray ones sodium, black ones carbon and white ones hydrogen atoms). Source data are provided as a Source data file. **c** Adsorption energies of hydrogen and ammonium in Na-PHI and H-PHI with respect to the number of heptazine units. $H_{ads} < 0$ means that adsorption is a spontaneous process, while desorption in uphill. When $H_{ads} > 0$—PHI does not bind hydrogen atom, while desorption is a spontaneous process.

material charging. Thus, in strongly reduced H-PHI and Na-PHI with up to 1e$^-$/1H$^+$ (or 1e$^-$/1NH$_4^+$) per 2 heptazine units $H_{ads} < 0$. In other words, desorption of e$^-$/H$^+$ (or e$^-$/NH$_4^+$) from such photocharged PHIs is endothermic process. On the other hand, in mildly-reduced PHIs with only 1e$^-$/1H$^+$ per 20 heptazine units $H_{ads} > 0$—desorption of e$^-$/H$^+$ is exothermic. Moreover, regardless of the degree of PHIs photocharging, desorption of e$^-$/NH$_4^+$ is always uphill. In the context of the benzylamine tetramerization cascade process, these results agree with the experiment in two aspects:

(1) Higher yield of **2a** is obtained, when the reaction is conducted at 80 °C compared to room temperature. Heating is required to overcome the energy barrier of PCET from photocharged H-PHI(e$^-$/H$^+$) to imine **1a** (Fig. 2).

(2) Higher yield of **2a** is obtained, when the reaction is conducted using a greater mass of g-CNs. Under such conditions materials are photocharged to lesser extent, which results in $H_{ads} > 0$ and weak

binding of e$^-$/H$^+$ by g-CN (e$^-$/H$^+$). Photocharged semiconductors can be divided into electronically doped and redox-shifted[42]. In the former, the quasi Fermi level of electrons moves to a more negative potential on the electrochemical scale. In the latter, the quasi Fermi level of electrons is equal to the potential of the conduction band. The dependence of $H_{ads}$ on the degree of g-CN photocharging and their reactivity toward oxidants suggest transition between these two classifications.

## Discussion

Taking into account the acquired experimental data and the results of DFT modeling, the mechanism of *aza*-pinacol coupling and a chemical model of structural changes in g-CN that occur in this process is shown in Fig. 7. Absorption of a photon by a g-CN gives an excited state, *g-CN(e$^-$/h$^+$), where e$^-$ denotes a photogenerated "hot" electron and h$^+$—a photogenerated hole (step i). According to the results of an EPR study,

**Fig. 7 | A scheme of *aza*-pinacol coupling mechanism.** *g-CN−electronically-excited state of carbon nitride, e⁻−photogenerated "hot" electron, h⁺−photogenerated hole, $e_t^-$−trapped electron, H⁺−proton, Im⁺− iminium cation derived from trialkylamine.

the distance between a hole and an electron in the charge separated state is ~2 nm, which corresponds to 4 heptazine units[43]. Quenching of the photogenerated hole by trialkylamine gives a photocharged state, in which negative charge is compensated by a proton and an iminium cation, g-CN($2e_t^-$/H⁺/Im⁺) (step ii). In this state, electrons are labeled as "$e_t^-$" to emphasize that they are trapped[44]. While the basicity of heptazine is low (p$K_a$ of the conjugated acid ~0 or negative), the heptazine radical anion is more basic. According to the results of DFT modeling, two heptazine units that form a "triangular pocket" at the edge plane of the graphitic structure pincer H⁺ and a charge-compensating iminium cation[45]. Due to these structural features the edge plane of g-CN is the site where oxidation of the substrate takes place. Hydrolysis of an iminium cation by trace amounts of water present in MeCN produces a molecule of dialkylamine, aldehyde and g-CN($2e_t^-$/2H⁺) (step iii). Taking into account the dependence of δ on the type of the tertiary amine, Et₃N vs. $^n$Bu₃N, the intermediary iminium cation remains bound to the surface of g-CN while its hydrolysis is slower compared to oxidation of the tertiary amine. In photocharged g-CN quenching experiments, however, electron transfer occurs from g-CN($2e_t^-$/2H⁺) to MV²⁺. Such assumption is also supported by the earlier observations that sorption of H⁺ at photocharged ZnO nanoparticles is more preferable compared to the bulky organic cations, such as $^n$Bu₄N⁺[46]. Experiments using photocharged carbon nitride semiconductors in the dark suggest that a trapped $e_t^-$ does not reduce **1a** to the benzylic radical **1a'**. Therefore, absorption of a second photon is necessary for the reaction to proceed (step iv). The excited state of the semiconductor is denoted as *g-CN(e⁻/h⁺/$2e_t^-$/2H⁺). Due to the potential of the valence band in Na-PHI and mpg-CN being +1.9 V and +1.2 V vs. SCE, respectively[47,48], the photogenerated hole reacts with $e_t^-$ (step v). The remaining in the conduction band "hot" electron (e⁻), which is not trapped, in combination with the proton stored at the edge plane of the graphitic

structure participates in PCET to yield **1a'** and g-CN($e_t^-$/H⁺) (step vi). As there is apparently no diastereoselective control, interaction between **1a'** and **1a** gives either *R,S*-**2a'** or *R,R*-**2a'**. Given that N-H BDFE in secondary amines is ~315–372 kJ mol⁻¹[49], *R,S*-**2a'** or *R,R*-**2a'** extract e⁻/H⁺ from g-CN($e_t^-$/H⁺), which produces *R,S*-**2a** or *R,R*-**2a**, respectively, and closes the photocatalytic cycle. When imine **1a** is no longer present in the reaction mixture, the reaction pathway shown in Fig. 7 is terminated at step iii, and leads to saturation of the g-CN structure with electrons, which is confirmed by kinetics studies (Supplementary Discussion 10). In a broader context, such behavior agrees well with the results of TiO₂ nanoparticles photocharging in *iso*-propanol/styrene oxide mixture[50] and ZnO in ethanol/acetaldehyde mixture[51]. Styrene oxide and acetaldehyde, similar to **1a**, act as e⁻/H⁺ acceptors lowering the number of electrons stored in the semiconductors. Tetramerization of benzylamine proceeds via a mechanism similar to *aza*-pinacol coupling although more elementary steps are involved (Supplementary Fig. 5).

We investigated the mechanism of the uptake of electrons and charge-compensating ions (H⁺, NH₄⁺) from amines by g-CN semiconductor upon irradiation with light, the storage of these electrons and cations in g-CN and the transfer of e⁻/H⁺ to the unsaturated substrate, such as an imine. The key findings of this study are:

1. Taking into account the dependence of the specific concentration of electrons (δ) on the type of tertiary amine, smaller Et₃N vs. larger $^n$Bu₃N, the results of DFT modeling and previous observations[41], we conclude that H⁺ and NH₄⁺ must be stored at the edge plane of g-CN nanocrystals in 'triangular pockets' created by two adjacent heptazine units.

2. δ is independent on the specific surface area (84–204 m² g⁻¹) of mpg-CN. This is explained by the fact that even for mpg-CN with relatively low surface area, 84 m² g⁻¹, in the photocharged state, ratio

between the number of surface heptazine units to the number of $e^-/H^+$ couples is >1.

3. DFT modeling revealed that the energy of cation adsorption ($H_{ads}$) depends on the degree of g-CN photocharging. In strongly reduced g-CN, with up to $1e^-$ per two heptazine units $H_{ads} < -10$ kJ mol$^{-1}$, i.e., the transfer of $e^-/H^+$ from such a g-CN state to a substrate is endergonic. In tetramerization of benzylamine induced by light irradiation, heating is therefore beneficial to facilitate $e^-/H^+$ transfer from reductively quenched g-CN to the intermediary imine. On the other hand, in mildly reduced g-CN with $1e^-$ per $\geq 4$ heptazine units, $H_{ads} > 0$, i.e., g-CN does not bind $e^-/H^+$, which makes $e^-/H^+$ transfer spontaneous. Substitution of $H^+$ by $NH_4^+$ in the micropore of PHI results in an even more substantial stabilizing influence onto the electron. H-PHI and Na-PHI photocharged in the presence of $NH_4^+$ are characterized by $H_{ads} < -70$ kJ mol$^{-1}$, which makes $e^-/H^+$ transfer to a substrate highly endergonic.

4. Overall, endergonic $e^-/H^+$ transfer from photocharged g-CN to a substrate could be enabled by (i) conducting the reaction in the presence of a larger amount of g-CN and (ii) increasing reaction temperature. In other words, materials carrying one extra electron per 1–3 heptazine units are weaker reductants due to $H_{ads} \ll 0$, while materials photocharged to the lowest degree, i.e., 1 e$^-$ per many heptazine units, are strong reductants due to $H_{ads} > 0$.

5. Exceptionally strong stabilization of electrons in PHIs was observed when photocharging was conducted in an aqueous solution of $NH_4COOH$. The photocharged PHI appears as a black solid and retains added electrons in air for hours. This property could be used to study such a state of g-CN under ambient conditions and to design air-stable hybrid composites.

## Methods

Additional experimental details are given in the Supplementary Information.

### Screening of reaction conditions of benzylamine tetramerization
A mixture of benzylamine (21 μL, 0.2 mmol), a semiconductor (20 mg), MeCN (2 mL) and a magnetic stirring bar were placed into a 5 mL screw-capped glass tube. The mixture was degassed three times using freeze-pump-thaw procedure and refilled with Ar. The mixture was stirred under irradiation with an LED for 24 h. After irradiation was stopped, a solution of 1,3,5-trimethoxybenzene (0.02 mmol, 0.1 mL, 0.2 M) in CH$_3$CN was added to the reaction mixture. The reaction mixture was transferred into a 2 mL Eppendorf tube and centrifuged (13,300 rpm, 11,000 × $g$, 3 min). The supernatant layer was separated. The semiconductor was washed with CH$_2$Cl$_2$ (3 × 2 mL) followed by centrifugation. The organic phases were combined in a flask and solvent was evaporated in vacuum (50 °C, 150 mbar). The residue was dissolved in CDCl$_3$ (0.9 mL) and analyzed by $^1$H NMR.

### Synthesis of R,S-2a-f and R,R-2a-f using mpg-CN-8nm-193
A mixture of imine **1** (3.5 mmol), mpg-CN-8nm-193 (0.7 g), MeCN (70 mL) and Et$_3$N (0.98 mL, 7 mmol) and a magnetic stirring bar were loaded into a three-neck glass tube equipped with a cold-finger, an inlet for N$_2$ and a temperature sensor that was connected to a thermostat. The mixture was purged with N$_2$ for 5 min and held under a positive pressure of N$_2$ of -0.1 bar to avoid possible air leakage into the reactor. The mixture was stirred under irradiation by blue LED (465 nm, 4.3 μmol s$^{-1}$) for 24 h. The temperature was set to 25 °C and controlled by a thermostat. The reaction progress was monitored by $^1$H NMR. Upon complete conversion of the imine **1**, the reaction mixture was concentrated in vacuum (+50 °C, 100 mbar). The resultant solid was washed with CH$_2$Cl$_2$ (4 × 8 mL) and separated by centrifugation (13,000 rpm, 11,000 × $g$, 3 min). Organic washings were combined and filtered through a 0.3 μm PTFE syringe filter. The solution was diluted

with CH$_2$Cl$_2$ to a total volume of 30 mL. An aliquot (0.5 mL) was mixed with a solution of internal standard, 1,3,5-trimethoxybenzene (0.02 mmol, 0.1 mL, 0.2 M in MeCN), and concentrated in vacuum (+50 °C, 100 mbar). The yield of R,S-**2** and R,R-**2** isomers was calculated from the $^1$H NMR spectra. The whole amount of the R,S-**2** and R,R-**2** isomers solution in CH$_2$Cl$_2$ was concentrated in vacuum (+50 °C, 50 mbar). The residue was triturated with pentane, the solid was filtered and washed with a small amount of pentane. Recrystallization from MTBE afforded R,S-**2**. The pentane solution was concentrated in vacuum (+50 °C, 500 mbar). The oily residue was purified by column chromatography using a stationary phase of basic Al$_2$O$_3$ (65 g). The R,R-**2** isomer was obtained by washing the column with equal volumes of eluent with gradually increasing polarity—(1) hexane 2 × 100 mL, (2) hexane:ethylacetate (19:1) 2 × 100 mL, (3) hexane:ethylacetate (9:1) 2 × 100 mL, (4) hexane:ethylacetate (4:1) 3 × 100 mL.

### Computational details
Density functional theory (DFT) calculations were conducted using the Gaussian and plane wave approach[52] as implemented in the CP2K suite of programs[53]. If not specified otherwise, all simulations were conducted using periodic boundary conditions and a plane wave cutoff of 500 Ry to represent the electron density, whereas the Kohn-Sham orbitals are described by a molecularly optimized double-zeta Gaussian basis set[46]. The Becke-based B97-3c exchange and correlation functional was employed in conjunction with an empirical dispersion correction to account for long-range van der Waals interactions[54]. Net atomic charges are quantified by means of the density derived electrostatic and chemical DDEC6 method[55].

## Data availability
The data that support the findings of this study are available from the corresponding author upon reasonable request. Source data are provided with this paper.

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

## Acknowledgements
The Max Planck Society is acknowledged for financial support of this project and providing infrastructure for research. Dr. Artem Mishchenko, Dr. Dmitry I. Sharapa, Dr. Clemens Schmitt, Michael Born, Jessica Brandt, Dr. Jiajia Cheng and Prof. Xinchen Wang are acknowledged for their help with the project.

## Author contributions
O.S. performed experimental studies, carried out the analysis and supervised the work. K.N. performed the computational studies. T.D.K. supervised the work. V.S. performed experimental studies. Y.M. performed experimental studies. M.A. supervised the work.

## Funding

## Competing interests
O.S. and M.A. declare personal financial interests. A patent WO/2019/081036 has been filed by the Max Planck Gesellschaft zur Förderung der Wissenschaften E.V. in which O.S. and M.A. are listed as co-authors. The remaining authors declare no competing interests.
