## [Peer Review File · Nature Communications]

Extent of Carbon Nitride Photocharging Controls Energetics of Hydrogen Transfer in Photochemical Cascade ProcessesREVIEWER COMMENTS

Reviewer #1 (Remarks to the Author):

This is an interesting manuscript. However, I believe it needs more attention before it can be considered for publication. Additionally, the work is rather specialised, and the manuscript would benefit if the generality of the findings could be (briefly) explained, so they were more obviously of interest to the readers of the journal.

Key results

The authors identify 3 objectives (i) identifying the locations in g-CN for the storage of charge compensating cations (in particular H⁺ and NH₄⁺); (ii) investigating the influence of H⁺ and NH₄⁺ on the reactivity of electrons trapped on g-CN; (iii) the extent to which micro and mesoporosity facilitate the trapping of electrons and the storage of H⁺ and NH₄⁺. They employ the photochemically induced tetramerization of benzylamine as both a model and a test reaction to investigate those objectives, and conduct DFT calculations.

However, the discussion in the conclusion section really addresses only the first of these objectives.

Validity

The preference of edge sites for storage of charge compensating cations as discussed on page 11 is quite speculative and not really supported by definitive wet chemistry evidence. The authors appear confident in making this claim because it is consistent with the results of the DFT calculations. They also conclude on page 15 that the edge plane is the site of oxidation based on the argument that iminium cations are too large to fit between the layers. On its own this argument is not compelling when there are many layered materials that swell to accommodate guests. It would be nice to get more convincing evidence of the location of charge compensating cations, e.g., if the numbers of charge compensating cations could be correlated with the relative numbers of edge sites.

The claim, on page 11 that the photochemical route is superior to the (single experiment describing the use of) Zn as a reductant may be true, but the claim as is, is hard to sustain given that not all photochemical experiments gave good conversions nor selectivities.

The authors discuss a mechanism for aza-pinacol coupling on page 15 and illustrate a catalytic cycle for the cascade reaction. The arguments would be strengthened if there were kinetic studies to support the proposed mechanism.

Significance

I enjoyed reading this manuscript – the concept of a cascade involving both an oxidation and a reduction, with reducing equivalents being “trapped”/stored between steps of the cascade is of undoubted interest. The manuscript reports an impressively large body of experimental data, (both in- and ex-silicio). The significance of the work would be increased by both relating the conclusions more strongly to the objectives and discussing similarities and contrasts with previous work.

Data and methodology

The data are comprehensive and the experimental work appears to have been conducted carefully. It is generally reported in sufficient detail that it could be reproduced at an independent laboratory. I suggest that the criteria for reversibility in the electrochemical experiments might be better described. There are some experiments that are discussed in the supporting information, but not in the manuscript

text. A minor point – the criteria for electrochemical reversibility could be stated in the SI.

Analytical approach

See above

Suggested improvements

See above

Clarity and context

I suggest that it would help the reader assimilate the large amount of data if the manuscript introduced the various experiments by indicating the hypothesis that was being tested (e.g., "in order to establish ... we did this" – followed by the experiment description and results). Although previous work is cited, its significance in relation to the present work might be explained more fully.

References

Both the manuscript and the SI are comprehensively referenced. However, the field is not large and the manuscript might reasonably describe comparisons with out studies, for example discussing whether the trapped electron/charge compensating cation model applies in other systems.

Reviewer #2 (Remarks to the Author):

In the article presented by Savateev et al, the authors describe photocharging properties and linked reactivity for oxidative-reductive cascade processes of two types of carbon nitrides (CN), namely mpg-CN and the two-dimensional poly(heptazine imide), which have distinct structures and electron storage capacities. They clarify important factors affecting the photocharging and discharging kinetics, as well as related energetics, which are of high importance for advancement of the field of photocharging materials in general, and their applications in photoredox chemistry as more special, but still highly important case. The multi-step photoredox processes described in the paper are well evidenced by numerous experiments. The energetic assumption made for explain different steps in the aza-pinacol coupling mechanism are further supported by DFT theory based modelling, increasing their plausibility, despite some open questions (vide infra). The key results are clearly summarized and the mechanisms supported by helpful illustrations. As such, I recommend the publication of this work in Nature Communications after some revisions.

1) The manuscript text is rather long and in some cases, it is difficult to follow why which reaction in the long tables in the main text and in the supporting information had been studied in which conditions and why – requiring more iterations. For enhanced readability to a broad audience, I would suggest a little more guidance through the importance of the tables and experiments, which could be done by more schematic figures, like Fig 7. This also affects the energetics of photocharging in different conditions, and the relation to proton adsorption (H_{ads}) energetics. It would probably be helpful to summarize this in a figure in the main text.

2) Photocharging experiments: When Na- or H-PHI is photocharged in presence of NH_4^+ , wouldn't a cation exchange lead to the transformation into (NH_4^+) -PHI, thereby modifying especially the properties of H-PHI? [ref 21, and Schlomberg et al, Chem. Mater. 2019, 31, 18, 7478–7486; Kroeger et al, Adv. Mater. 2022, 34, 2107061] Can it be proven that H-PHI does not form an ionic structure? The band gap

of H-PHI is typically larger than for ionic PHIs eg (see citations above)

3) The authors describe the distance of electrons to holes to be 4nm – is this more an excitonic, or a charge separated state?

4) Heating described to facilitate e⁻/H⁺ transfer due the apparently endergonic nature of this process in some conditions. Generally, that fits in the line of thought, but only if the heating does not affect the materials charge storage capacity (denoted by delta). Does heating to 80C affect the energetics storage capacity of the materials? You heat after photocharging. A control experiment with photocharging at 80C should be done here. If the properties are affected (and especially, if the capacity is increased), how can this be disentangled from reactivity and energy barriers? Also, why is there no data with Na-PHI (RT) a higher mass than 20mg in Fig 2a, to compare the effects of heating in more detail for this material? (black open circles)

5) Photocharging stability with NH₄COOH after air exposure and subsequent discussion (page 9 and 10): The long term stability in air intriguing and very interesting for further exploration. Can you elaborate more quantitatively how many hours the mixture is stable? Also, the pictures in Fig 4 suggest severe agglomeration or flocculation. Might it be that the accessibility of the material by dissolved oxygen is reduced, e.g. either due to changes in the oxygen diffusion properties in the solution, or at the surface of (clogged) agglomerates? Some indications in this direction would be helpful to make use of such effects in future. And an optical characterization (eg absorbance/UV-VIS spectrum) of this black state should be provided.

6) Computation: Fig 6: color code for nitrogen and sodium not well chosen making it difficult to read. Also, Na-PHI should have a negatively charged backbone, while Na⁺ is residing in pores, correct? Where are the negative charges located balancing the Na cation? And how come the solvent can be omitted in these calculations? The ions appear to reside in solvated structures (e.g. Schlomberg Chem Sci; Kroeger Adv Mat, see above). Would the solvation shells (especially with water) not affect the cation positions, charge storage and H adsorption energetics significantly, also due to steric hindering?

7) There are some typos in the texts, and the grammar of some sentences seems not very common. I suggest one more round of careful proof-reading.

With kind regards,

Filip Podjaski

Reviewer #3 (Remarks to the Author):

This manuscript describes the energetics of g-CN discharging. They have investigated the transfer of e⁻/H⁺ from g-CN photo charged with electrons and protons (H⁺)/(NH₄⁺) ions to an oxidant. The finding reveals that NH₄⁺ exerts a robust stabilizing effect, which makes e⁻/H⁺ transfer uphill. In aqueous conditions, NH₄⁺ forms stable photo-charged sodium poly(heptazine imide). However, the mildly reduced g-CN, H⁺ do not stabilize electrons, which results in the spontaneous transfer of e⁻/H⁺ to oxidants. Facile transfer of e⁻/H⁺ is a key step in oxidative-reductive cascade – tetramerization of

benzylic amines, which is a two-step process viz. i) oxidation of two benzylic amine molecules to the imine with associated with storage of $2e^-/2H^+$ in g-CN and ii) reduction of the imine to α -aminoalkyl radical involving $1e^-/1H^+$ transfer.

They have demonstrated the tetramerization of benzylic amines via a cascade process where several steps are merged together. The reaction has been carefully performed, the diastereomeric ratio has been determined by 1H NMR. Each and every step of this reaction was thoroughly assessed. The scope of the aza-pinacol coupling was then extended to several other benzylic systems. The experimental findings have been supported through DFT calculation. Finally, by taking into the experimental observation and DFT calculations acceptable mechanism has been proposed.

Considering the importance of g-CN this manuscript can be accepted for publications

Reviewer #4 (Remarks to the Author):

In this study, the authors investigated the photocatalytic activity of two different g-CNs with varying pore sizes. They rationally selected the tetramerization of benzylic amines as the target cascade reaction and achieved high yields through electron stabilization via coupling with NH_4^+ . Taking the photocatalytic oxidative-reductive cascade-tetramerization of benzylic amines as the example, the authors revealed that g-CN played a dual-function on the oxidative-reductive cascade, oxidizing benzylamine to form imine 1a through storing hydrogen and simultaneously reducing imine 1a to R,S-2a and R,R-2a through transferring the stored hydrogen.

They further studied how the degree of carbon nitride photocharging influenced on the e^-/H^+ transfer during the reaction, which significantly determined the yield of the final products. Also, they found the protons (H^+) and ammonium cations (NH_4^+) showed a distinct difference in terms of their cation adsorption energy, leading to different reducing capacity. Finally, the authors presented a detailed and self-consistent description of their findings by combing experiments with the DFT simulation. I

recommend to accept this manuscript for publication after a revision by addressing the following issues:

1. In Figure 3c, the authors discuss the specific concentration of electrons (δ) and its strong correlation with the benzylamine per carbon nitride mass. However, it appears that δ is an intrinsic property, independent of the mass of semiconductors. Is it possible that the decrease in visible light irradiance per particle with increasing semiconductor concentration in the solution is the reason behind this observation? Additionally, could the value of δ vary depending on the design (thickness) of the glass vial?
2. The results regarding the yield based on the degree of electron stabilization, as compared by the adsorption energies of NH_4^+ and H^+ , are very interesting. Considering other adsorbed cations in addition to NH_4^+ , it is anticipated that an optimal yield could be achieved by identifying an energy position using hydrogen adsorption energy as a descriptor. Is there any other cation that could be considered for this operation?
3. In Table 1, entry 4, it is observed that NH_4^+ does not work on mpg-CN. Could the authors please provide a discussion comparing the adsorption energy of NH_4^+ on mpg-CN to that on PHI? This would help in understanding the underlying reasons for this discrepancy.

4. On page 3, line 14, the sentence reads, "Conversion of benzylamine, yield of 1a, and combined yield of R,S- and R,R-2a diastereomers versus the mass of carbon nitride semiconductor, Na-PHI and mpg-CN, are plotted in Figure 2a." However, Figure 2 is not labeled separately as a and b. Please ensure that the labeling in the paragraph and figures is consistent.

Point-by-point responses to referees' and editorials comments to the manuscript "**Degree of Carbon Nitride Photocharging Controls Energetics of Hydrogen Transfer in Photochemical Cascade Processes**" (NCOMMS-23-15960).

Referees' and editor's comments

Our responses

Changes made in the manuscript

REVIEWER COMMENTS

Reviewer #1 (Remarks to the Author):

This is an interesting manuscript. However, I believe it needs more attention before it can be considered for publication. Additionally, the work is rather specialised, and the manuscript would benefit if the generality of the findings could be (briefly) explained, so they were more obviously of interest to the readers of the journal.

Response: We are pleased to know that referee found our work interesting and thank for the constructive comments, which we addressed below. Especially we would like to thank the referee for the time dedicated to correct grammar and indicate typos.

Key results

The authors identify 3 objectives (i) identifying the locations in g-CN for the storage of charge compensating cations (in particular H⁺ and NH₄⁺); (ii) investigating the influence of H⁺ and NH₄⁺ on the reactivity of electrons trapped on g-CN; (iii) the extent to which micro and mesoporosity facilitate the trapping of electrons and the storage of H⁺ and NH₄⁺. They employ the photochemically induced tetramerization of benzylamine as both a model and a test reaction to investigate those objectives, and conduct DFT calculations.

However, the discussion in the conclusion section really addresses only the first of these objectives.

Response: The section was corrected to cover all three objectives

We investigated the mechanism of the uptake of electrons and charge-compensating ions (H⁺, NH₄⁺) from amines by g-CN semiconductor upon irradiation with light, the storage of these

electrons and cations in g-CN and the transfer of e^-/H^+ to the unsaturated substrate, such as an imine. The key findings of this study are:

1. Taking into account the dependence of the specific concentration of electrons (δ) on the type of tertiary amine, smaller Et_3N vs. larger nBu_3N , the results of DFT modeling and previous observations,¹ we conclude that H^+ and NH_4^+ must be stored at the edge plane of g-CN nanocrystals in 'triangular pockets' created by two adjacent heptazine units.

2. δ is independent on the specific surface area ($84-204\text{ m}^2\text{ g}^{-1}$) of mpg-CN. This is explained by the fact that even for mpg-CN with relatively low surface area, $84\text{ m}^2\text{ g}^{-1}$, in the photocharged state, ratio between the number of surface heptazine units to the number of e^-/H^+ couples is > 1.

3. DFT modeling revealed that the energy of cation adsorption (H_{ads}) depends on the degree of g-CN photocharging. In strongly reduced g-CN, with up to $1e^-$ per two heptazine units $H_{ads} < -10\text{ kJ mol}^{-1}$, i.e. the transfer of e^-/H^+ from such a g-CN state to a substrate is endergonic. In tetramerization of benzylamine induced by light irradiation, heating is therefore beneficial to facilitate e^-/H^+ transfer from reductively quenched g-CN to the intermediary imine. On the other hand, in mildly reduced g-CN with $1e^-$ per ≥ 4 heptazine units, $H_{ads} > 0$, i.e. g-CN does not bind e^-/H^+ , which makes e^-/H^+ transfer spontaneous. Substitution of H^+ by NH_4^+ in the micropore of PHI results in an even more substantial stabilizing influence onto the electron. H-PHI and Na-PHI photocharged in the presence of NH_4^+ are characterized by $H_{ads} < -70\text{ kJ mol}^{-1}$, which makes e^-/H^+ transfer to a substrate highly endergonic.

4. Overall, endergonic e^-/H^+ transfer from photocharged g-CN to a substrate could be enabled by *i)* conducting the reaction in the presence of a larger amount of g-CN and *ii)* increasing reaction temperature. In other words, materials carrying one extra electron per 1-3 heptazine units are weaker reductants due to $H_{ads} \ll 0$, while materials photocharged to the lowest degree, i.e. $1e^-$ per many heptazine units, are strong reductants due to $H_{ads} > 0$.

5. Exceptionally strong stabilization of electrons in PHIs was observed when photocharging was conducted in an aqueous solution of NH_4COOH . The photocharged PHI appears as a black solid and retains added electrons in air for hours. This property could be used to study such a state of g-CN under ambient conditions and to design air-stable hybrid composites.

Validity

The preference of edge sites for storage of charge compensating cations as discussed on page 11 is quite speculative and not really supported by definitive wet chemistry evidence. The authors appear confident in making this claim because it is consistent with the results of the DFT calculations. They also conclude on page 15 that the edge plane is the site of oxidation based on the argument that iminium cations are too large to fit between the layers. On its own this argument is not compelling when there are many layered materials that swell to accommodate guests. It would be nice to get more convincing evidence of the location of

charge compensating cations, e.g., if the numbers of charge compensating cations could be correlated with the relative numbers of edge sites.

Response: The following section was added as Supplementary Discussion 7.

Considering the specific surface area of g-CN materials used in the photocharging experiments and the obtained corresponding δ values, we calculated the ratio of heptazine units per electron. In the case of mpg-CN with specific surface area in the range 84-204 m² g⁻¹, statistically one electron is stored per 1-7 heptazine units (electron donor Et₃N) or 2-9 heptazine units (electron donor ⁿBu₃N, Supplementary Table 33). These results do not exclude the possibility that in mpg-CN electrons and charge-compensating cations are stored exclusively on the material surface. However, in case of Na-PHI due to its lower S_{SA} of only 1 m² g⁻¹ and higher δ values compared to mpg-CN, up to 167 or 111 electrons (Et₃N and ⁿBu₃N electron donors respectively) must be stored in a single heptazine unit. Assuming full reduction of a heptazine unit, i.e., disintegration of the material with the formation of three CH₄ and four NH₃ molecules, this process would allow for storage of only 24 electrons in the products of heptazine unit reduction. Therefore, in poly(heptazine imides), electrons and charge compensating cations, must be stored in the micropores. The surface of micropores on the interface with the reaction medium is composed exclusively of (ac) and (bc) crystal cell facets – both are edge planes.

On the other hand, assuming transport of electrons into the bulk of materials accompanied by intercalation of charge compensating cations between the layers, one electron is stored per 14-30 heptazine units. In principle, such a possibility could not be excluded for both types of g-CN materials. However, due to microporous structure ion transport into the bulk of Na-PHI should be more facile compared to mpg-CN.²

Taking into account results published earlier and general pathway of tertiary amines oxidation,^{3,4,5} the mechanism of aza-pinacol coupling implies formation of the iminium cation (Figure 7). A photocharged mpg-CN with charge-compensating iminium cations, g-CN(2e⁻/H⁺/Im⁺), is a possible intermediate. However, given high reactivity of the iminium cation toward water, which is unavoidably present in MeCN and/or adsorbed on the surface of carbon nitrides, it is converted into dialkylamine and aldehyde. These two species were detected by ¹H NMR, while the released H⁺ becomes the charge-compensating ion instead of the iminium cation for the stored electron.

The claim, on page 11 that the photochemical route is superior to the (single experiment describing the use of) Zn as a reductant may be true, but the claim as is, is hard to sustain given that not all photochemical experiments gave good conversions nor selectivities.

Response: Discussion was adjusted.

These results point at the advantage of the photochemical approach compared to chemical reduction with Zn dust to reach higher selectivity toward **2a**, 38% vs. 66%.

Imidazolidine is a by-product detected in the reaction mixture (Supplementary Figure 3). This compound is formed from **2c** and acetaldehyde, which was generated in situ upon triethylamine oxidation.⁶

Supplementary Fig. 3. Structure of imidazolidine that is formed upon **2c** condensation with acetaldehyde.

The authors discuss a mechanism for aza-pinacol coupling on page 15 and illustrate a catalytic cycle for the cascade reaction. The arguments would be strengthened if there were kinetic studies to support the proposed mechanism.

Response: As suggested by referee, in order to support the proposed mechanism, we conducted kinetic study, in which we monitored 1) conversion of imine **1a** and Et_3N , 2) yield of the coupling product **2a** and Et_2NH , and 3) specific concentration of electrons accumulated in mpg-CN. Given the high sensitivity of photocharged mpg-CN towards O_2 , we designed a setup, which allows taking aliquots from the reactor and quenching photocharged mpg-CN with MV^{2+} followed by measuring the absorption at 605 nm under O_2 -free conditions. The setup is shown in Supplementary Figure 12.

Supplementary Fig. 12. Schematic representation of the setup and photographs of its elements that was used to perform kinetic measurements studies. Absolute pressure values are shown.

Description of the reaction mixture sampling to determine conversion/yield of organic compounds and specific concentration of electrons in mpg-CN was added to the Supporting Information.

Supplementary Discussion 10

The results of kinetic studies are summarized in the following points:

1. At the beginning of the reaction, the ratio between *R,S-2a* and *R,R-2a* is > 1 , which is in agreement with the experiments conducted in batch (Supplementary Fig. 13). Due to *R,S-2a* low solubility in MeCN, the ratio between *R,S-* and *R,R-2a* becomes noticeably smaller than unity when imine **1a** conversion reaches 72% (reaction mixture irradiation for 15 h). After kinetic study was completed, the setup was disassembled. Crystals of *R,S-2a* were observed to deposit on the cooler. Therefore, the measured by ^1H NMR combined yield of the coupling product is lower than the actual.

2. Before the irradiation of the reaction mixture was initiated, 65% of the initial amount of triethylamine was adsorbed on the surface of mpg-CN. Given electron-deficient character of heptazine-based carbon nitride, such results are not surprising. We speculate that the driving force for sorption of electron-rich Et_3N on the surface of mpg-CN is the formation of charge-transfer complex.

3. Although conversion of Et_3N reached 95% after 43 h, the yield of Et_2NH was only 18% suggesting that large fraction remains adsorbed on the surface of mpg-CN.

4. When imine **1a** conversion is $< 30\%$, δ is relatively low – electrons are not accumulated at the significant extent in mpg-CN. However, photocharging of mpg-CN is accelerated when imine **1a** conversion is $> 70\%$ (Supplementary Fig. 14).

The results of kinetic studies are shown in Supplementary Figure 13 and 14, and Supplementary Table 30.

Supplementary Fig. 13. Results of kinetics study.

Supplementary Fig. 14. Dependence of δ on 1a conversion. These data points were acquired at 0, 169 \pm 5, 197 \pm 5, 902 \pm 3, 1172 \pm 4, 1816 \pm 9, \pm 2550 \pm 9 min and taken from Supplementary Table 30.

Analysis of the kinetics study results allowed to make the following conclusions, which was added as Supplementary Discussion 10:

The results of kinetic studies are summarized in the following points:

1. At the beginning of the reaction, the ratio between *R,S*-**2a** and *R,R*-**2a** is > 1 , which is in agreement with the experiments conducted in batch (Supplementary Fig. 13). Due to *R,S*-**2a** low solubility in MeCN, the ratio between *R,S*- and *R,R*-**2a** becomes noticeably smaller than unity when imine **1a** conversion reaches 72% (reaction mixture irradiation for 15 h). After kinetic study was completed, the setup was disassembled. Crystals of *R,S*-**2a** were observed to deposit on the cooler. Therefore, the measured by ^1H NMR combined yield of the coupling product is lower than the actual.

2. Before the irradiation of the reaction mixture was initiated, 65% of the initial amount of triethylamine was adsorbed on the surface of mpg-CN. Given electron-deficient character of heptazine-based carbon nitride, such results are not surprising. We speculate that the driving force for sorption of electron-rich Et_3N on the surface of mpg-CN is the formation of charge-transfer complex.

3. Although conversion of Et_3N reached 95% after 43 h, the yield of Et_2NH was only 18% suggesting that large fraction remains adsorbed on the surface of mpg-CN.

4. When imine **1a** conversion is < 30%, δ is relatively low – electrons are not accumulated at the significant extent in mpg-CN. However, photocharging of mpg-CN is accelerated when imine **1a** conversion is > 70% (Supplementary Fig. 14).

Significance

I enjoyed reading this manuscript – the concept of a cascade involving both an oxidation and a reduction, with reducing equivalents being “trapped”/stored between steps of the cascade is of undoubted interest. The manuscript reports an impressively large body of experimental data, (both in- and ex-silicio). The significance of the work would be increased by both relating the conclusions more strongly to the objectives and discussing similarities and contrasts with previous work.

Response: Conclusion section was rephrased to make to more relevant to the objectives. The present results were discussed in relation to previous works.

1. The results of kinetics study that δ scales with imine **1a** conversion agree with the results of TiO₂ nanoparticles photocharging in *iso*-propanol/styrene oxide mixture⁷ and ZnO nanoparticles photocharging in ethanol/acetaldehyde mixture.⁸ Styrene oxide and acetaldehyde, similar to imine **1a**, act as e⁻/H⁺ acceptors lowering the number of electrons stored in semiconductor nanoparticles. In *aza*-pinacol coupling, over the reaction time mpg-CN accumulates greater number of electrons when concentration of imine **1a** decreases, while ethanediamine **2a** cannot act as oxidant.

When imine **1a** is no longer present in the reaction mixture, the reaction pathway shown in Figure 7 is terminated at step *iii*, and leads to saturation of the g-CN structure with electrons, which is confirmed by kinetics studies (Supplementary Discussion 10). In a broader context, such behavior agrees well with the results of TiO₂ nanoparticles photocharging in *iso*-propanol/styrene oxide mixture⁷ and ZnO in ethanol/acetaldehyde mixture.⁸ Styrene oxide and acetaldehyde, similar to **1a**, act as e⁻/H⁺ acceptors lowering the number of electrons stored in the semiconductors.

2. In the presence of several kind of cations in the reaction mixture, sorption of H⁺ by photocharged ZnO nanoparticles is more preferable compared bulky organic cations, such as ⁿBu₄N⁺.⁹ Due to smaller radius of H⁺ compared to the iminium cation and sensitivity of the latter to hydrolysis, H⁺ are the most likely counterions in the photocharged carbon nitrides.

In photocharged g-CN quenching experiments, however, electron transfer occurs from g-CN(2e⁻/2H⁺) to MV²⁺. Such assumption is also supported by the earlier observations that sorption of H⁺ at photocharged ZnO nanoparticles is more preferable compared to the bulky organic cations, such as ⁿBu₄N⁺.⁹

Data and methodology

The data are comprehensive and the experimental work appears to have been conducted carefully. It is generally reported in sufficient detail that it could be reproduced at an independent laboratory. I suggest that the criteria for reversibility in the electrochemical experiments might be better described. There are some experiments that are discussed in the supporting information, but not in the manuscript text. A minor point – the criteria for electrochemical reversibility could be stated in the SI.

Response: Supplementary Figure 6 and Discussion 8 were added

Supplementary Fig. 6. CV curves of $MV^{2+} 2PF_6^-$, Et_3N , nBu_3N and $PhCH_2NH_2$ in 0.1 M solution of $nBu_4N^+ ClO_4^-$ in MeCN. Scan rate 50 $mV s^{-1}$.

Supplementary Discussion 8

$MV^{2+} 2PF_6^-$ in MeCN undergoes electrochemically reversible one-electron reduction to the radical cation, $MV^{•+} PF_6^-$, at the potential $-0.83 V$ vs. Fc^+/Fc followed by electrochemically reversible one-electron reduction to the dimethylbipyridinylidene at the potential $-1.24 V$ vs. Fc^+/Fc (Supplementary Fig. 6). Both processes are electrochemically reversible because peak separations in both cases are approx. 60 mV.¹⁰

Tertiary amines, Et_3N and nBu_3N , undergo electrochemically and chemically irreversible one-electron oxidation to the corresponding radical cations at the potentials $+0.34$ and $+0.30 V$ vs. Fc^+/Fc respectively. Because the reduction peak is absent in the CV curve upon scanning back from positive to negative potential, these processes are electrochemically irreversible. In addition, the radical cations of tertiary amines are short-lived species and undergo further spontaneous transformations, such as cleavage of $(sp^3)C-H$ bond next to nitrogen atom, with the formation of the corresponding iminium cation.¹¹ The lifetime of the tertiary amine radical cation is significantly shorter compared to the duration of the CV experiment. In other words, at the scan rate $50 mV s^{-1}$ used to determine oxidation potentials of the tertiary amines, it is impossible

to detect electron transfer from the working electrode to the radical cation, because the latter is decomposed and therefore absent in the proximity of the electrode.

Analytical approach

See above

Suggested improvements

See above

Clarity and context

I suggest that it would help the reader assimilate the large amount of data if the manuscript introduced the various experiments by indicating the hypothesis that was being tested (e.g., "in order to establish we did this" – followed by the experiment description and results). Although previous work is cited, it's significance in relation to the present work might be explained more fully.

Response: Discussion was adjusted to explain logic behind conducting specific experiments. The significance of previous works in relation to this work was explained more fully.

When imine **1a** is no longer present in the reaction mixture, the reaction pathway shown in Figure 7 is terminated at step *iii*, and leads to saturation of the g-CN structure with electrons, which is confirmed by kinetics studies (Supplementary Discussion 10). In a broader context, such behavior agrees well with the results of TiO₂ nanoparticles photocharging in *iso*-propanol/styrene oxide mixture⁷ and ZnO in ethanol/acetaldehyde mixture.⁸ Styrene oxide and acetaldehyde, similar to **1a**, act as e⁻/H⁺ acceptors lowering the number of electrons stored in the semiconductors.

In photocharged g-CN quenching experiments, however, electron transfer occurs from g-CN(2e⁻/2H⁺) to MV²⁺. Such assumption is also supported by the earlier observations that sorption of H⁺ at photocharged ZnO nanoparticles is more preferable compared to the bulky organic cations, such as ⁿBu₄N⁺.⁹

References

Both the manuscript and the SI are comprehensively referenced. However, the field is not large and the manuscript might reasonably describe comparisons with out studies, for example discussing whether the trapped electron/charge compensating cation model applies in other systems.

Response: In the presence of several kind of cations in the reaction mixture, sorption of H^+ by photocharged ZnO nanoparticles is more preferable compared bulky organic cations, such as ${}^nBu_4N^+$.⁹ Due to smaller radius of H^+ compared to the iminium cation and sensitivity of the latter to hydrolysis, H^+ are the most likely counterions in the photocharged carbon nitrides.

Discussion was adjusted to show generality of the results. After publishing these results, the measured specific concentration of electrons stored in carbon nitrides will be used to extend the Database of Photocharged Materials (see for example, <https://public.tableau.com/app/profile/oleksandr.savatieiev> and Savateev, O. Adv. Energy Mater. 2022, 12, 2200352.). In this way, quantitative parameters determined in this work may be considered in relation with other photocharged materials.

Reviewer #2 (Remarks to the Author):

In the article presented by Savateev et al, the authors describe photocharging properties and linked reactivity for oxidative-reductive cascade processes of two types of carbon nitrides (CN), namely mpg-CN and the two-dimensional poly(heptazine imide), which have distinct structures and electron storage capacities. They clarify important factors affecting the photocharging and discharging kinetics, as well as related energetics, which are of high importance for advancement of the field of photocharging materials in general, and their applications in photoredox chemistry as more special, but still highly important case. The multi-step photoredox processes described in the paper are well evidenced by numerous experiments. The energetic assumption made for explain different steps in the aza-pinacol coupling mechanism are further supported by DFT theory based modelling, increasing their plausibility, despite some open questions (vide infra). The key results are clearly summarized and the mechanisms supported by helpful illustrations. As such, I recommend the publication of this work in Nature Communications after some revisions.

Response: Dear Dr. Filip Podjaski, thank you very much for your constructive comments, which we addressed point-by-point below.

1) The manuscript text is rather long and in some cases, it is difficult to follow why which reaction in the long tables in the main text and in the supporting information had been studied in which conditions and why – requiring more iterations. For enhanced readability to a broad audience, I would suggest a little more guidance through the importance of the tables and experiments, which could be done by more schematic figures, like Fig 7. This also affects the energetics of photocharging in different conditions, and the relation to proton adsorption (H_{ads}) energetics. It would probably be helpful to summarize this in a figure in the main text.

Response: Discussion was adjusted to clarify logic behind conducting specific experiments.

2) Photocharging experiments: When Na- or H-PHI is photocharged in presence of NH₄⁺, wouldn't a cation exchange lead to the transformation into (NH₄⁺)-PHI, thereby modifying especially the properties of H-PHI? [ref 21, and Schlomberg et al, Chem. Mater. 2019, 31, 18, 7478–7486; Kroeger et al, Adv. Mater. 2022, 34, 2107061]

Response: The results are shown in Supplementary Discussion 4.

Supplementary Discussion 4

To check if in benzylamine tetramerization experiments conducted at room temperature and at 80°C, substitution of Na in Na-PHI by other cations took place, samples (original mass of Na-PHI 80 mg) after washing with CH₂Cl₂ were dispersed in deionized water (1.5 mL). Na-PHI was separated by centrifugation at 13000 rpm. The liquid phase was decanted and solid was dried in vacuum (+60°C, 20 mbar). Fresh Na-PHI, and recovered samples were characterized by DRUV-vis absorption and FT-IR spectroscopies. DRUV-vis spectra revealed that optical band gap of Na-PHI did not change, although additional states were introduced, which are observed as absorption in the range 470-650 nm (Supplementary Fig. 15a). FT-IR spectra of three materials are similar (Supplementary Fig. 15b). The peak at approx. 985 nm, which is assigned to N-Na vibrations,¹² is also present in three samples suggesting that extensive substitution of Na⁺ by NH₄⁺ or H⁺ did not take place.

Next, we checked if there is any correlation between the amount of Na⁺ extracted from Na-PHI and the amount of electrons stored in Na-PHI. Na content in Na-PHI was determined using inductively coupled plasma optical emission spectroscopy (ICP-OES). The dependence of n_{Na^+} on n_{e^-} is shown in Supplementary Fig. 16 (original data is shown in Supplementary Table 34). Assuming that there is a linear relationship between n_{Na^+} and n_{e^-} , which is expressed by the function $n_{\text{Na}^+} = k \times n_{e^-} + b$, slopes were determined to be $k = 25.3$ (reaction conducted at 80°C, $R^2 = 0.544$) and $k = 7.1$ (reaction conducted at room temperature, $R^2 = 0.999$). In other words, storage of one electron in Na-PHI occurs at the expense of approx. 25 and 7 Na⁺ cations. To maintain charge balance, Na⁺ ions that were extracted from Na-PHI are like to be replaced by NH₄⁺ and H⁺. While NH₄⁺ ions are generated upon coupling of two benzylamine molecules, H⁺ might originate from water that is present in MeCN, crystallization water in Na-PHI or water adsorbed on Na-PHI. This process may be described by the equation:

We speculate that the reason that storage of one electron in Na-PHI requires extraction of more than one Na⁺ is better stabilization of the stored electrons in the bulk of the material. Taking into account ordered transport of ions through Na-PHI microchannels,² in order for electron and charge-compensating cation (NH₄⁺ or H⁺) to reach the bulk up to 7 and 25 Na⁺ must leave the micropore when photocharging is conducted at room temperature and 80°C, respectively. Overall, such mechanism agrees with the proposed earlier.²

Furthermore, the fitting lines intersect X-axis at 1.3 and 2.7 μmol , respectively. In other words, storage of up to 2.7 μmol electrons in Na-PHI should be possible without extracting Na^+ . We speculate that such mode of electron storage is possible when occurs exclusively on the outer surface of Na-PHI particles without perturbing the environment of the micropores. Taking into account specific surface area of Na-PHI used to create Supplementary Fig. 16 ($1 \text{ m}^2 \text{ g}^{-1}$), unit cell parameters ($a = b = 1.238 \text{ nm}$, $c = 0.33 \text{ nm}$),¹³ the corresponding areas of the unit cell individual facets ($S_{ab} = 1.533 \text{ nm}^2$, $S_{ac} = S_{bc} = 0.409 \text{ nm}^2$) and the number of heptazine units per unit cell projected onto the specific cell facet ($k = 2$, see Supplementary Fig. 2 for crystal structure visualization), the amount of heptazine units on the surface of Na-PHI particles is 8, 2 or 8 $\mu\text{mol g}^{-1}$, when the surface is terminated by (ac), (ab) or (bc) facets, respectively. Na-PHI loading greater than 1.5-6.2 g (photocharging at 80°C) and 0.74-3.0 g (photocharging at room temperature) would be required to store 1.3-2.7 μmol of electrons and charge-compensating ions exclusively on the outer surface of Na-PHI particles. However, such mode of electron storage is very unlikely to be achieved due to better stabilization of electrons in the bulk of the material (see a paragraph above). Therefore, it is a theoretical value.

Supplementary Fig. 15. **Characterization of Na-PHI recovered after benzylamine tetramerization.** Reaction was conducted at room temperature and at 80°C. Na-PHI mass taken for the reaction 80 mg. **a** DRUV-vis spectra. **b** FT-IR spectra.

Supplementary Table 34. Correlation of the number of electrons accumulated in Na-PHI with the number of Na^+ extracted from Na-PHI.

$m(\text{Na-PHI})$, $\text{mg}^{[a]}$	t , $^{\circ}\text{C}^{[b]}$	$\omega(\text{Na})$, wt. $\%^{[c]}$	n_{Na^+} , $\mu\text{mol}^{[d]}$	δ , $\mu\text{mol g}^{-1[e]}$	n_{e^-} , $\mu\text{mol}^{[f]}$
5	25±5	12.2±0.48	3	644	3.2
10	25±5	10.3±2.26	15	470	4.7
20	25±5	10.2±0.61	30	353	7.1

40	25±5	10.2±3.16	61	282	11.2
80	25±5	10.6±1.06	108	—	—
120	25±5	11.2±1.76	130	—	—
160	25±5	11.4±0.3	160	—	—
5	80	11.0±2.79	6	445	2.2
10	80	9.41±1.06	19	202	2.0
20	80	9.66±0.79	35	106	2.1
40	80	10.2±0.61	61	92	3.7
80	80	11.3±1.36	83	—	—
120	80	11.9±0.81	94	—	—
160	80	12.0±1.26	118	—	—

[a] Mass of Na-PHI ($S_{S.A.}$ 1 m² g⁻¹) taken to perform benzylamine tetramerization.

[b] Temperature of benzylamine tetramerization reaction.

[c] Na content in Na-PHI after recovery from the reaction mixture. Determined by ICP-OES. Na content in fresh Na-PHI is $\omega(\text{Na}_{ref}) = 13.7 \pm 1.67$ wt. % (mean±std., n = 3).

[d] The amount of Na⁺ (μmol) extracted from Na-PHI was calculated according to the equation:

$$n_{\text{Na}^+} = \frac{(\omega(\text{Na}_{ref}) - \omega(\text{Na})) \cdot 10^{-2} \cdot m(\text{Na-PHI}) \cdot 10^{-3} \cdot 10^6}{A_{\text{Na}}} \quad (15)$$

where $\omega(\text{Na}_{ref})$ – Na content in fresh Na-PHI, $\omega(\text{Na}_{ref}) = 13.7 \pm 1.67$ wt. % (mean±std, n = 3). $\omega(\text{Na})$ – Na content in Na-PHI recovered from the reaction mixture, wt. %. $m(\text{Na-PHI})$ – mass of Na-PHI taken to perform benzylamine tetramerization, mg. A_{Na} – atomic mass of sodium, $A_{\text{Na}} = 23$ g mol⁻¹.

[e] Specific concentration of electrons (δ) in Na-PHI that was photocharged in the presence of benzylamine at room temperature or at 80°C. Data were taken from Supplementary Table 4.

[f] The amount of electrons stored in Na-PHI (mol) was calculated according to the equation:

$$n_{e^-} = \delta \cdot 10^{-6} \cdot m(\text{Na-PHI}) \cdot 10^{-3} \quad (16)$$

Supplementary Fig. 10. Dependence of benzylamine conversion on the surface area of the semiconductor.

Can it be proven that H-PHI does not form an ionic structure? The band gap of H-PHI is typically larger than for ionic PHIs eg (see citations above)

Response: In this work, we prepared H-PHI by Na-PHI treatment with HCl. Visually H-PHI appears as pale-yellow solid suggesting wider optical band gap compared to Na-PHI. In photocharging experiments using benzylamine as electron donor, the obtained δ values are similar to that of mpg-CN (see Figure 3c).

Figure 3. **Photocharging of g-CN semiconductors.** **a** UV-vis absorption spectra of mpg-CN dispersion in $i\text{Pr}_2\text{NEt}$ (0.1 M) solution in MeCN prior to irradiation with light (black), after irradiation (blue) and after exposure of photocharged mpg-CN to air (gray). **b** Photographs of the reaction mixture (mpg-CN-8nm-193 20 mg) before light irradiation and after irradiation at 465 nm for 24 h before exposure to air. **c** Dependence of δ (avg \pm std, $n = 3$) on mass of the semiconductor photocharged at room temperature and 80°C. Deviation of the data point (avg \pm std, $n = 5$) obtained for Na-PHI (1 mg) from the trend is due to low optical density of the reaction mixture.

Coupling of two benzylamine molecules is accompanied by the formation of NH_4^+ . Substitution of H^+ by alkali metal ion occurs in aqueous environment at $\text{pH} > 7$.^{12,2,14} Benzylamine tetramerization was conducted in less polar solvent, MeCN, without adjusting the pH. Therefore, we speculate that under such conditions spontaneous H^+ -to- NH_4^+ ion exchange in H-PHI does not occur. However, given that NH_4^+ serves as the counter ion to the stored electron (see below), such ion exchange might proceed via material photocharging.

3) The authors describe the distance of electrons to holes to be 4nm – is this more an excitonic, or a charge separated state?

Response: In the reference article, it is not explicitly specified if such a distance corresponds to excitonic or charge-separated state. However, given the timescale of EPR study, i.e. > 200 ns, it must be charge-separated state. Discussion was adjusted:

According to the results of an EPR study, the distance between a hole and an electron in presumably charge separated state is ~2 nm, which corresponds to 4 heptazine units.¹⁵

4) Heating described to facilitate e⁻/H⁺ transfer due the apparently endergonic nature of this process in some conditions. Generally, that fits in the line of thought, but only if the heating does not affect the materials charge storage capacity (denoted by delta). Does heating to 80C affect the energetics storage capacity of the materials?

You heat after photocharging. A control experiment with photocharging at 80C should be done here. If the properties are affected (and especially, if the capacity is increased), how can this be disentangled from reactivity and energy barriers?

Response: Discussion was adjusted.

Na-PHI shows systematically lower δ values, when photocharging is conducted at 80°C suggesting that it is an endergonic process (see the results of DFT calculations below and Supplementary Discussion 3).

Supplementary Discussion 3

We performed photocharging of Na-PHI at 80°C using benzylamine as electron donor. Quenching with MV²⁺ 2PF₆⁻ was performed at room temperature. The corresponding δ values were plotted vs. Na-PHI mass in Figure 3c and original data was added to Supplementary Table 1. Na-PHI shows systematically lower δ values, when photocharging was conducted at 80°C compared to that at room temperature, which agrees with the results of DFT modelling. Given that sorption of e⁻/H⁺ is exergonic, while desorption of e⁻/H⁺ is therefore endergonic, heating of the reaction mixture favors discharging.

Heating facilitates dispersion of Na-PHI, which results in colloidal solution of higher optical density. Therefore, even at low Na-PHI loading (1 mg), absorption of light by the colloidal solution was improved compared to the same amount of Na-PHI photocharged at room temperature. Therefore, comparable δ values were obtained for 1 mg of Na-PHI photocharged at room temperature and at 80°C.

Figure 3. **Photocharging of g-CN semiconductors.** **a** UV-vis absorption spectra of mp-g-CN dispersion in *i*Pr₂NEt (0.1 M) solution in MeCN prior to irradiation with light (black), after irradiation (blue) and after exposure of photocharged mp-g-CN to air (gray). **b** Photographs of the reaction mixture (mpg-CN-8nm-193 20 mg) before light irradiation and after irradiation at 465 nm for 24 h before exposure to air. **c** Dependence of δ (avg \pm std, $n = 3$) on mass of the semiconductor photocharged at room temperature and 80°C. Deviation of the data point (avg \pm std, $n = 5$) obtained for Na-PHI (1 mg) from the trend is due to low optical density of the reaction mixture.

Also, why is there no data with Na-PHI (RT) a higher mass than 20mg in Fig 2a, to compare the effects of heating in more detail for this material? (black open circles)

Response: In an attempt to add missing values to Figure 2a, we found that the batch of Na-PHI that was originally used to create this figure was completely consumed. Figure 2a was recreated by repeating a series of experiments using a new batch of Na-PHI. We found that the new batch of Na-PHI gave significantly lower conversion/yield. The reason is due to the lower specific surface area of the new batch ($1 \text{ m}^2 \text{ g}^{-1}$) compared to the old batch ($35 \text{ m}^2 \text{ g}^{-1}$) determined from N₂ physisorption. Comparison of performance of these two Na-PHI batches in benzylamine tetramerization is given in Supplementary Figure 9. Note that δ values reported in Figure 3c and 4b were obtained using Na-PHI with low specific surface area ($1 \text{ m}^2 \text{ g}^{-1}$).

Supplementary Fig. 9. Benzylamine tetramerization mediated by Na-PHI. **a** Na-PHI with S_{SA} $35 \text{ m}^2 \text{ g}^{-1}$. **b** Na-PHI with S_{SA} $1 \text{ m}^2 \text{ g}^{-1}$.

Figure 2. Dependence of benzylamine conversion and yield of imine **1a**, *R,S*- and *R,R*-**2a** on mass of semiconductors and reaction temperature. Filled symbols correspond to yield, empty – to conversion. Gray/black colors correspond to experiments conducted at room temperature, blue

– at 80°C. Conditions: benzylamine (0.2 mmol), MeCN (2 mL), Ar, 24 h, light 410 nm (39 ± 8 mW cm^{-2}). Error bars denote standard deviation ($n = 3$, experiments conducted in parallel).

5) Photocharging stability with NH_4COOH after air exposure and subsequent discussion (page 9 and 10): The long term stability in air intriguing and very interesting for further exploration. Can you elaborate more quantitatively how many hours the mixture is stable?

Also, the pictures in Fig 4 suggest severe agglomeration or flocculation. Might it be that the accessibility of the material by dissolved oxygen is reduced, e.g. either due to changes in the oxygen diffusion properties in the solution, or at the surface of (clogged) agglomerates?

Response: Supplementary Discussion 5 was added.

Supplementary Discussion 5

To determine more accurately how long Na-PHI can remain photocharged in air, we conducted two experiments. In both experiments, a mixture of Na-PHI and NH_4COOH in MeCN/ H_2O (3:1) was irradiated at 410 nm for 12 h upon stirring. After that the reaction mixture in the first flask was allowed to react with O_2 from air upon stirring. MeCN/ H_2O from the second flask was decanted and the residue was dried in vacuum (room temperature, 0.2 mbar) for 30 min and exposed to air. Change of Na-PHI color in both cases is shown in Supplementary Fig. 11.

It is obvious that in the first case, oxidation of Na-PHI occurs fast (within 30 min), which is due to improved mass transport of O_2 to Na-PHI particles. Oxidation of photocharged Na-PHI in the solid state proceeds slower – some areas of Na-PHI film retained its gray metallic sheen even after 4 h.

Flocculation observed in these experiments is due to separation of MeCN/ H_2O phases. While these two solvents are immiscible, phase separation occurs in the presence of electrolyte, such as NH_4COOH , which decreases solubility of MeCN in water. Black droplets are rich in water, Na-PHI and NH_4COOH .

Supplementary Fig. 11. Oxidation of Na-PHI photocharged using NH_4COOH . a Reaction mixture was exposed to air. **b** Solution was decanted followed by drying the residue in vacuum and exposure to air.

Some indications in this direction would be helpful to make use of such effects in future. And an optical characterization (eg absorbance/UV-VIS spectrum) of this black state should be provided.

Response: Supplementary Discussion 6 was added.

Supplementary Discussion 6

A mixture of Na-PHI, NH_4COOH in MeCN/ H_2O was photocharged upon irradiation at 410 nm. Within 5 s of mixture exposure to light, it changed color from yellow to dark blue. This color is due to appearance of an additional absorption band stretching from 450 nm to nIR (Supplementary Fig. 7). It should be noted that such short exposure of Na-PHI to light did not affect its intrinsic band gap – a band in the UV-vis range of the electromagnetic spectrum with the onset of absorption at approx. 450 nm is observed in pristine Na-PHI and Na-PHI exposed to light for 5 s.

Extended irradiation of the mixture resulted in further color change to dark gray. This color transition is due to broadening of the absorption peak at approx. 650 nm – the resultant material absorbs all photons in the UV-nIR range.

Supplementary Fig. 7. DRUV-vis spectra of Na-PHI photocharged using NH_4COOH . Irradiation at 410 nm, 4 W optical power, photon flux $14 \mu\text{mol s}^{-1}$.

6) Computation: Fig 6: color code for nitrogen and sodium not well chosen making it difficult to read.

Response: Color of sodium was changed to gray.

a

b

c

Also, Na-PHI should have a negatively charged backbone, while Na⁺ is residing in pores, correct? Where are the negative charges located balancing the Na cation? And how come the solvent can be omitted in these calculations? The ions appear to reside in solvated structures (e.g. Schlomberg Chem Sci; Kroeger Adv Mat, see above). Would the solvation shells (especially with water) not affect the cation positions, charge storage and H adsorption energetics significantly, also due to steric hindering?

Response: Right, the Polyheptazine network acts as a negative counterpart towards the sodium ions within the pore, yet the charge is not equally distributed. It is mainly located on those nitrogen atoms located on the outer part of the heptazine (N2, N3, N4, see Supplementary Table 26 and 27). As Supplementary Table 28 shows, adding an explicit solvent (here: water) affects the

charge distribution, however the overall effect is rather small and not of qualitative nature. Furthermore, adding multiple solvent molecules increases the amount of degrees of freedom, making it challenging to find a global minimum. Also, the interaction of the solvent with the PHI disguises the interaction of the adsorbate with the backbone. As Supplementary Table 29 shows, the average cation positions are influenced and the distance to the PHI increases, due to the steric hindrance you already mentioned. The effect of the solvent gets significant when looking at the adsorption energies in Supplementary Table 19. While the released adsorption energy of the small hydrogen atom increases, due to hydrogen bonding with the solvent while not being bothered by the decreased free space. Ammonium is hindered sterically, but also benefits energetically from interacting with the solvent. However, these energies resemble more of a snapshot strongly dependent on the current solvent configuration and less influenced by the e-density within the PHI material.

Supplementary Table 26. Charges of atoms in Na-PHI.

	Arithmetic mean Charge, ^[a] [e]	Standard Deviation, [e]
N1	-0.215	0.001
N2	-0.592	0.017
N3	-0.584	0.033
N4	-0.547	0.017
C1	0.562	0.023
C2	0.512	0.008
Na	0.885	0.01

^[a] Charges were calculated for 20-PHI systems via DDEC6. Atom Labels are depicted in Supplementary Figure 17.

Supplementary Table 27. Charges of atoms in H-PHI.

	Arithmetic mean Charge, ^[a] [e]	Standard Deviation, [e]
N1	-0.257	0.005
N2	-0.507	0.020
N3	-0.479	0.017
N4	-0.370	0.003
C1	0.552	0.017
C2	0.529	0.018

^[a] Charges were calculated for 20-PHI systems via DDEC6. Atom Labels are depicted in Supplementary Figure 17.

Supplementary Table 28. Charges of atoms in Na-PHI solvated with 5-H₂O per Na atom.

	Arithmetic mean Charge, ^[a] [e]	Standard Deviation, [e]
N1	-0.231	0.014
N2	-0.526	0.044

N3	-0.519	0.023
N4	-0.485	0.056
C1	0.568	0.036
C2	0.539	0.016
Na	0.911	0.012

^[a] Charges were calculated for 20-PHI systems via DDEC6. Atom Labels are depicted in Supplementary Figure 17.

Supplementary Table 29. Distance Na-Backbone in Na-PHI.

	Average distance [Å]	Standard deviation
Na-N2 distance	2.64	0.096
Na-N2 distance (with water)	2.85	0.552
Na-N3 distance	2.41	0.080
Na-N3 distance (with water)	3.12	0.556

Supplementary Table 19. Adsorption energies in solvated Na-PHI.

Structure	Adsorption energy, kJ mol ⁻¹
H+/2-Na-PHI	-32.29
H+/4-Na-PHI	-69.30
H+/8-Na-PHI	-65.54
H+/20-Na-PHI	0.62
NH4/2-Na-PHI	117.42
NH4/4-Na-PHI	63.78
NH4/8-Na-PHI	-24.40
NH4/20-Na-PHI	117.56

These findings were summarized in the Supporting Information file:

Supplementary Discussion 9

Atomic charges in PHIs. Having a closer look at the charges of Na-PHI gives further insights. They were obtained using DDEC6 as implemented in Chargemol, with the cube-files calculated via CP2K as described in the manuscript, although an increased cutoff of 1000 Ry has been used. This is necessary for an appropriate assignment of the electrons. To have the charges in an experimental feasible and computational accurate manner only two decimals were evaluated. The Supplementary Table 24 and 25 give the average charge in [e] for all atoms of each element in the structure, with its standard deviation, if applicable. Overall, the charges of the elements do not drastically vary with changing system size. Although a slight increase in positive charge by circa 0.02 e is observable for carbon and sodium from 2 to 20 heptazine units. This is

accompanied by an increase of negative charge on the bridging nitrogen that forms a covalent bond with a hydrogen during the adsorption process.

In general, the nitrogen atoms on the outer part of the heptazine imide cumulate most of the negative charge, which is ascribable to the charge transfer with sodium or other interacting atoms/molecules, but also to their lone pair. Charges on individual atoms in PHIs composed of 20 heptazine units are summarized in Supplementary Table 26 (Na-PHI) and Supplementary Table 27 (H-PHI). Assignment of labels to specific atoms in PHI is given in Supplementary Fig. 17. The trends are identical in the smaller systems, deviations are only small (in the range of $0.02e$ compared to 2 heptazine PHIs).

When Na-PHI without solvent (Supplementary Table 27) and with solvent (Supplementary Table 28) are compared, charges vary in second decimal due to interaction with the solvent, e.g. hydrogen bonding. Overall influence is of quantitative nature and rather small.

Na-backbone distances in PHIs. Average distance of sodium with CN-Backbone increases when adding water (Supplementary Table 29), but the actual values are again highly dependent on the position and orientation of the explicit solvent. There is a shift of the sodium towards the pore center, as one can derive from the Na-N3 distance (Supplementary Fig. 17). With water molecules solvating the ion the higher distance is sterically favorable.

Adsorption energies in solvated PHIs. PHI systems with each sodium ion explicitly solvated by 5 water molecules², were investigated regarding the adsorption energies of H^+/NH_4^+ . These energies are shown in Supplementary Table 19. There is no noticeable trend, and the energies are rather dominated by the interaction with the solvent. These interactions are strongly influenced by small differences in the orientation and position of the molecules which result from the geometric optimization of the system. Omitting the solvent reveals more information about the interaction of the adsorbate with the adsorbent in regard of differently charged systems, as can be seen in Fig. 6.

7) There are some typos in the texts, and the grammar of some sentences seems not very common. I suggest one more round of careful proof-reading.

Response: The manuscript was revised by all co-authors. The grammar was corrected by referee #1. All these suggestions were applied.

With kind regards,

Filip Podjaski

Reviewer #3 (Remarks to the Author):

This manuscript describes the energetics of g-CN discharging. They have investigated the transfer of e^-/H^+ from g-CN photo charged with electrons and protons ($H^+/(NH_4^+)$) ions to an

oxidant. The finding reveals that NH_4^+ exerts a robust stabilizing effect, which makes e^-/H^+ transfer uphill. In aqueous conditions, NH_4^+ forms stable photo-charged sodium poly(heptazine imide). However, the mildly reduced g-CN, H^+ do not stabilize electrons, which results in the spontaneous transfer of e^-/H^+ to oxidants. Facile transfer of e^-/H^+ is a key step in oxidative-reductive cascade – tetramerization of benzylic amines, which is a two-step process viz. i) oxidation of two benzylic amine molecules to the imine with associated with storage of $2e^-/2\text{H}^+$ in g-CN and ii) reduction of the imine to α -aminoalkyl radical involving $1e^-/1\text{H}^+$ transfer.

They have demonstrated the tetramerization of benzylic amines via a cascade process where several steps are merged together. The reaction has been carefully performed, the diastereomeric ratio has been determined by ^1H NMR. Each and every step of this reaction was thoroughly assessed. The scope of the aza-pinacol coupling was then extended to several other benzylic systems. The experimental findings have been supported through DFT calculation. Finally, by taking into the experimental observation and DFT calculations acceptable mechanism has been proposed.

Considering the importance of g-CN this manuscript can be accepted for publications
Response: We appreciate referees' positive feedback and suggestion to accept our manuscript for publication.

Reviewer #4 (Remarks to the Author):

In this study, the authors investigated the photocatalytic activity of two different g-CN with varying pore sizes. They rationally selected the tetramerization of benzylic amines as the target cascade reaction and achieved high yields through electron stabilization via coupling with NH_4^+ . Taking the photocatalytic oxidative-reductive cascade-tetramerization of benzylic amines as the example, the authors revealed that g-CN played a dual-function on the oxidative-reductive cascade, oxidizing benzylamine to form imine 1a through storing hydrogen and simultaneously reducing imine 1a to R,S-2a and R,R-2a through transferring the stored hydrogen.

They further studied how the degree of carbon nitride photocharging influenced on the e^-/H^+ transfer during the reaction, which significantly determined the yield of the final products. Also, they found the protons (H^+) and ammonium cations (NH_4^+) showed a distinct difference in terms of their cation adsorption energy, leading to different reducing capacity. Finally, the authors presented a detailed and self-consistent description of their findings by combing experiments with the DFT simulation. I recommend to accept this manuscript for publication after a revision by addressing the following issues:

Response: We appreciate referee's positive feedback and constructive comments that we addressed.

1. In Figure 3c, the authors discuss the specific concentration of electrons (δ) and its strong correlation with the benzylamine per carbon nitride mass. However, it appears that δ is an intrinsic property, independent of the mass of semiconductors. Is it possible that the decrease in visible light irradiance per particle with increasing semiconductor concentration in the solution is the reason behind this observation?

Response: Supplementary Discussion 2 was added.

Supplementary Discussion 2

To check if higher δ values are obtained, when greater excess of benzylamine per carbon nitride mass is used, we conducted photocharging of Na-PHI (5 mg, i.e., the same optical density) in the presence of various benzylamine concentrations. The results are shown in Supplementary Fig. 8. Note that δ value obtained using 0.3 mol L⁻¹ benzylamine solution in MeCN might be overestimated due to spontaneous reduction of MV²⁺ 2PF₆⁻ by benzylamine in the dark as indicated in the Supplementary Methods section "Measurement of the number of stored in semiconductor electrons". Overall, δ values are independent of benzylamine concentration in the range 0.01-0.2 mol L⁻¹, i.e., δ is independent on carbon nitride to benzylamine ratio.

Supplementary Fig. 8. Dependence of δ on concentration of benzylamine. Data points for K-PHI are taken from reference ¹⁶.

Additionally, could the value of δ vary depending on the design (thickness) of the glass vial?

Response: The following part was added to the Supplementary Discussion 2.

According to Bunsen-Roscoe law the amount of converted reagent proportional to the amount absorbed light. Therefore, using thick vials or vials which are not transparent to photons of specific wavelength will give lower δ values. In this work, we used vials that absorb approx. 10% of incident photons at $\lambda > 350$ nm.

Earlier it was reported that saturation of carbon nitride materials with electrons occurs within 2 h of irradiation, i.e., δ reaches its maximum value under given conditions.¹⁷ Taking into account time of a typical photocharging experiment (24 h), during which reaction the mixture received portion of photons approx. 0.2 mol (410 nm, 39 mW cm⁻², vial outer surface area 18 cm²) the apparent quantum yield (AQY) that is defined by the number of electrons stored in carbon nitride to the number of incident photons is approx. 2×10^{-5} (or 0.002%). Kinetic studies suggest that amines have strong affinity to carbon nitride. Therefore, we speculate that their oxidation is fast. Relatively low AQY value indicates that in the typical photocharging experiment, the semiconductor receives large excess of photons. Overall, these analyzes imply that the reported δ values are maximum under the specific conditions, i.e., carbon nitrides are saturated with electrons.

2. The results regarding the yield based on the degree of electron stabilization, as compared by the adsorption energies of NH₄⁺ and H⁺, are very interesting. Considering other adsorbed cations in addition to NH₄⁺, it is anticipated that an optimal yield could be achieved by identifying an energy position using hydrogen adsorption energy as a descriptor. Is there any other cation that could be considered for this operation?

Response: In tetramerization of benzylamine, two ammonium cations and two electrons are released and stored in Na-PHI. Given that sorption of NH₄⁺ at Na-PHI is exergonic, these cations substitute other cations – we have shown this by measuring by ICP-OES Na content in Na-PHI recovered after the reaction. In other words, adding other electrolyte into the reaction mixture

in attempt to facilitate hydrogen atom transfer from photocharged poly(heptazine imide) to imine **1a** most likely would not affect the performance of carbon nitride semiconductor, i.e., yield of ethanediamines **2a**.

From a practical standpoint, it is possible to consider other derivatives of toluene as reaction partners with benzylamines in synthesis of ethanediamines, which would release other cations different from NH_4^+ . These precursors, for example, are trialkylbenzylammonium and dialkylbenzylsulfonium salts. Especially using trimethylbenzylammonium salts, formation of dibenzylamine is expected in dark.¹⁸

Analysis of possible reaction intermediates suggests that the first step of the cascade process, i.e. synthesis of imine, is eliminated. Instead, nucleophilic attack of benzylamine at ammonium or sulfonium center in dark gives dibenzyl amine. Therefore, the role of carbon nitride sensitizer is to enable dimerization of dibenzylamine.¹⁹ Trialkylammonium cation and proton will compete for the adsorption on carbon nitride surface. Given the weak basicity of dialkylsulfides, in case of using dialkylbenzylsulfonium salts, the only cation adsorbed on Na-PHI surface would be H⁺.

3. In Table 1, entry 4, it is observed that NH₄⁺ does not work on mpg-CN. Could the authors

please provide a discussion comparing the adsorption energy of NH_4^+ on mpg-CN to that on PHI? This would help in understanding the underlying reasons for this discrepancy.

Response: Photocharging of H-PHI using benzylamine as electron donor gave δ values, which are similar to that of mpg-CN – please note that in Figure 3c gray line (H-PHI) follows the black one (mpg-CN) almost perfectly.

These results imply that the energetics of NH_4^+ adsorption on H-PHI is similar to mpg-CN. For this reason, we used the report earlier H-PHI crystal structure to perform DFT modeling.

4. On page 3, line 14, the sentence reads, "Conversion of benzylamine, yield of 1a, and combined yield of R,S- and R,R-2a diastereomers versus the mass of carbon nitride semiconductor, Na-PHI and mpg-CN, are plotted in Figure 2a." However, Figure 2 is not labeled separately as a and b. Please ensure that the labeling in the paragraph and figures is consistent.

Response: Thank you very much for indicating the typo! It was corrected.

Conversion of benzylamine, the yield of 1a and the combined yield of R,S- and R,R-2a diastereomers versus the mass of carbon nitride semiconductor, Na-PHI and mpg-CN, are plotted in Figure 2.

References that are included in the point-by-point response file, manuscript and Supplementary Information file:

1. Lin L., *et al.* Molecular-level insights on the reactive facet of carbon nitride single crystals photocatalysing overall water splitting. *Nature Catalysis* **3**, 649-655 (2020).

2. Kröger J., *et al.* Conductivity Mechanism in Ionic 2D Carbon Nitrides: From Hydrated Ion Motion to Enhanced Photocatalysis. *Advanced Materials* **34**, 2107061 (2022).
3. Savateev O. & Zou Y. Identification of the Structure of Triethanolamine Oxygenation Products in Carbon Nitride Photocatalysis. *ChemistryOpen* **11**, e202200095 (2022).
4. Ghosh T., Das A. & König B. Photocatalytic N-formylation of amines via a reductive quenching cycle in the presence of air. *Organic & Biomolecular Chemistry* **15**, 2536-2540 (2017).
5. Wada Y., Kitamura T. & Yanagida S. CO₂-fixation into organic carbonyl compounds in visible-light-induced photocatalysis of linear aromatic compounds. *Research on Chemical Intermediates* **26**, 153-159 (2000).
6. Ando S., Xiao B. & Ishizuka T. Synthesis of Imidazolinium Salts by Pd/C-Catalyzed Dehydrogenation of Imidazolidines. *Eur J Org Chem* **2021**, 4551-4554 (2021).
7. Li Y., Ji H., Chen C., Ma W. & Zhao J. Concerted Two-Electron Transfer and High Selectivity of TiO₂ in Photocatalyzed Deoxygenation of Epoxides. *Angewandte Chemie International Edition* **52**, 12636-12640 (2013).
8. Carroll G. M., Schimpf A. M., Tsui E. Y. & Gamelin D. R. Redox Potentials of Colloidal n-Type ZnO Nanocrystals: Effects of Confinement, Electron Density, and Fermi-Level Pinning by Aldehyde Hydrogenation. *Journal of the American Chemical Society* **137**, 11163-11169 (2015).
9. Valdez C. N., Delley M. F. & Mayer J. M. Cation Effects on the Reduction of Colloidal ZnO Nanocrystals. *Journal of the American Chemical Society* **140**, 8924-8933 (2018).
10. Cook S. K. & Horrocks B. R. Heterogeneous Electron-Transfer Rates for the Reduction of Viologen Derivatives at Platinum and Bismuth Electrodes in Acetonitrile. *ChemElectroChem* **4**, 320-331 (2017).
11. Hu J., Wang J., Nguyen T. H. & Zheng N. The chemistry of amine radical cations produced by visible light photoredox catalysis. *Beilstein Journal of Organic Chemistry* **9**, 1977-2001 (2013).
12. Savateev A., Pronkin S., Willinger M. G., Antonietti M. & Dontsova D. Towards Organic Zeolites and Inclusion Catalysts: Heptazine Imide Salts Can Exchange Metal Cations in the Solid State. *Chemistry – An Asian Journal* **12**, 1517-1522 (2017).
13. Chen Z., *et al.* “The Easier the Better” Preparation of Efficient Photocatalysts—Metastable Poly(heptazine imide) Salts. *Advanced Materials* **29**, 1700555 (2017).

14. Podjaski F., Kröger J. & Lotsch B. V. Toward an Aqueous Solar Battery: Direct Electrochemical Storage of Solar Energy in Carbon Nitrides. *Adv Mater* **30**, 1705477 (2018).
15. Actis A., et al. Morphology and Light-Dependent Spatial Distribution of Spin Defects in Carbon Nitride. *Angewandte Chemie International Edition* **61**, e202210640 (2022).
16. Markushyna Y., et al. Green radicals of potassium poly(heptazine imide) using light and benzylamine. *Journal of Materials Chemistry A* **7**, 24771-24775 (2019).
17. Lau V. W.-h., et al. Dark Photocatalysis: Storage of Solar Energy in Carbon Nitride for Time-Delayed Hydrogen Generation. *Angewandte Chemie International Edition* **56**, 510-514 (2017).
18. Lawrence S. A. Substitution on the Amine Nitrogen. In: *Category 5, Compounds with One Saturated Carbon Heteroatom Bond*. 1st Edition edn. Georg Thieme Verlag (2009).
19. Mitkina T., Stanglmair C., Setzer W., Gruber M., Kisch H. & König B. Visible light mediated homo- and heterocoupling of benzyl alcohols and benzyl amines on polycrystalline cadmium sulfide. *Organic & Biomolecular Chemistry* **10**, 3556-3561 (2012).

Sincerely,

Oleksandr Savateev

REVIEWERS' COMMENTS

Reviewer #1 (Remarks to the Author):

The authors seem to have responded appropriately to the referees' comments. There are still some typos (e.g., tree CH₄) and unconventional grammatical constructions, but these can be addressed during subsequent processing, if appropriate. Importantly, questions regarding the science have been addressed. Subject to the comments of other referees. The authors are to be commended for agreeing to conduct additional experiments. I believe this version meets the criteria for publication in the journal (albeit that the topic is somewhat specialized).

Reviewer #2 (Remarks to the Author):

The authors have made large efforts to address the comments and question I had, as well as those of the other reviewers. This happened with care and in great detail. The manuscript has improved much in clarity and ambiguity, while maintaining its significance. The shift to and addition of supplementary discussions is helpful for specialized readers to now better understand all considerations. I especially appreciate the control experiments added to clarify issues raised, such as the kinetic study done for the important question of reviewer 1 and the one on reactivity with oxygen from myself, the possibility of ion exchange from Na-PHI with NH₄, and the improved data and clarity on the numerical simulations. Also, the readability has improved due a clearer discussion structure. As such, I recommend the publication of this manuscript, which does not require any further changes.

However, I would have appreciated if my questions on ion exchange by NH₄ were also tested on H-PHI on which I had emphasized more (both aspects of question 2), forming possibly an NH₄-PHI structure more easily than the Na-PHI case discussed.

To elaborate in more detail: The authors write now that they “speculate that under such conditions spontaneous H⁺-to-NH₄⁺ ion exchange in H-PHI does not occur.” This is of course not wrong to do, but it also could have been verified. The authors argue that cation exchange was reported to work in alkaline solutions at pH>7 as cited (refs 2,12 and 14 in rebuttal), but none of the publications seems to say that this was exclusive and must be limited to strongly alkaline solutions. It seems likely that these more (slightly) alkaline conditions favour only the kinetics of the exchange, without being a necessary thermodynamic argument. The conditions in ref 2 (Kröger et al, *Advanced Materials* 34, 2107061 (2022)) are in fact only mildly alkaline (pH9-10). And the main author himself writes in ref 12 (Savateev et al, *Chemistry – An Asian Journal* 12, 1517-1522 (2017)) that: “For example, the original PHI-K can be reconstituted by agitating PHI-Mg in a concentrated KCl solution.” – which does not appear alkaline to me (typically slightly acidic, pH~6 due to ambient CO₂ contact e.g.).

Also, for Figure 4b, such an experiment would have potentially added a nice and perhaps important datapoint. Here, the authors indicate changes in photocharging capacity Δ when photocharging is conducted in presence of NH₄PF₆ (indicated by arrows showing a change in Δ_{max}), which indeed increases for Na-PHI in all cases, but the case is omitted for H-PHI.

With kind regards,
F. Podjaski

Reviewer #4 (Remarks to the Author):

My concerns have been addressed in the revisions and the revised manuscript is acceptable for publication.

Point-by-point responses to referees' and editorials comments to the manuscript "Extent of Carbon Nitride Photocharging Controls Energetics of Hydrogen Transfer in Photochemical Cascade Processes" (NCOMMS-23-15960A).

Referees' comments

Our responses

REVIEWER COMMENTS

Reviewer #1 (Remarks to the Author):

The authors seem to have responded appropriately to the referees' comments. There are still some typos (e.g., tree CH₄) and unconventional grammatical constructions, but these can be addressed during subsequent processing, if appropriate. Importantly, questions regarding the science have been addressed. Subject to the comments of other referees. The authors are to be commended for agreeing to conduct additional experiments. I believe this version meets the criteria for publication in the journal (albeit that the topic is somewhat specialized).
Response: We appreciate referee's positive feedback and recommendation to publish our work in Nature Communications.

Reviewer #2 (Remarks to the Author):

The authors have made large efforts to address the comments and question I had, as well as those of the other reviewers. This happened with care and in great detail. The manuscript has improved much in clarity and ambiguity, while maintaining its significance. The shift to and addition of supplementary discussions is helpful for specialized readers to now better understand all considerations. I especially appreciate the control experiments added to clarify issues raised, such as the kinetic study done for the important question of reviewer 1 and the one on reactivity with oxygen from myself, the possibility of ion exchange from Na-PHI with NH₄, and the improved data and clarity on the numerical simulations. Also, the readability has improved due a clearer discussion structure. As such, I recommend the publication of this manuscript, which does not require any further changes.

However, I would have appreciated if my questions on ion exchange by NH₄ were also tested on H-PHI on which I had emphasized more (both aspects of question 2), forming possibly an NH₄-PHI structure more easily than the Na-PHI case discussed. To elaborate in more detail: The authors write now that they "speculate that under such conditions spontaneous H⁺-to-NH₄⁺

ion exchange in H-PHI does not occur.” This is of course not wrong to do, but it also could have been verified. The authors argue that cation exchange was reported to work in alkaline solutions at $\text{pH} > 7$ as cited (refs 2,12 and 14 in rebuttal), but none of the publications seems to say that this was exclusive and must be limited to strongly alkaline solutions. It seems likely that these more (slightly) alkaline conditions favour only the kinetics of the exchange, without being a necessary thermodynamic argument. The conditions in ref 2 (Kröger et al, *Advanced Materials* 34, 2107061 (2022)) are in fact only mildly alkaline ($\text{pH} 9-10$). And the main author himself writes in ref 12 (Savateev et al, *Chemistry – An Asian Journal* 12, 1517-1522 (2017)) that: “For example, the original PHI-K can be reconstituted by agitating PHI-Mg in a concentrated KCl solution.” – which does not appear alkaline to me (typically slightly acidic, $\text{pH} \sim 6$ due to ambient CO_2 contact e.g.).

Also, for Figure 4b, such an experiment would have potentially added a nice and perhaps important datapoint. Here, the authors indicate changes in photocharging capacity Δ when photocharging is conducted in presence of NH_4PF_6 (indicated by arrows showing a change in Δ_{max}), which indeed increases for Na-PHI in all cases, but the case is omitted for H-PHI.

With kind regards,
F. Podjaski

Response: Dear Dr. Podjaski, thank you very much for your positive evaluation of the revised version of the manuscript and your recommendation to publish it in *Nature Communications*.

We agree that exchange of H^+ by NH_4^+ in H-PHI under the photocatalytic conditions is important for fundamental understanding of the process. In literature, ion exchange in PHIs was investigated in aqueous solution, while in our work, most of photocatalytic experiments and all measurements of Δ values were conducted in anhydrous acetonitrile. We agree that strongly alkaline environment is not necessary for successful ion exchange (at least in aqueous solution). However, due to lower polarity of MeCN compared to water, H^+ -to- NH_4^+ exchange might be less favorable in anhydrous environment. To conclude whether or not exchange of H^+ by NH_4^+ occurs in H-PHI when dispersed in non-aqueous environment, substantial amount of experimental work is required. It will be a topic of the independent publication in the future.

Reviewer #4 (Remarks to the Author):

My concerns have been addressed in the revisions and the revised manuscript is acceptable for publication.

Response: We appreciate referee’s positive feedback and the recommendation to accept the article for publication in *Nature Communications*.

Sincerely,
Oleksandr Savateev

25.10.2023, Hong Kong